# A moisture function of soil heterotrophic respiration that incorporates microscale processes

Zhifeng Yan[1], Ben Bond-Lamberty[2], Katherine E. Todd-Brown[3], Vanessa L. Bailey[3], SiLiang Li[1], CongQiang Liu[4] & Chongxuan Liu[5]

Soil heterotrophic respiration (HR) is an important source of soil-to-atmosphere $CO_2$ flux, but its response to changes in soil water content ($\theta$) is poorly understood. Earth system models commonly use empirical moisture functions to describe the HR–$\theta$ relationship, introducing significant uncertainty in predicting $CO_2$ flux from soils. Generalized, mechanistic models that address this uncertainty are thus urgently needed. Here we derive, test, and calibrate a novel moisture function, $f_m$, that encapsulates primary physicochemical and biological processes controlling soil HR. We validated $f_m$ using simulation results and published experimental data, and established the quantitative relationships between parameters of $f_m$ and measurable soil properties, which enables $f_m$ to predict the HR–$\theta$ relationships for different soils across spatial scales. The $f_m$ function predicted comparable HR–$\theta$ relationships with laboratory and field measurements, and may reduce the uncertainty in predicting the response of soil organic carbon stocks to climate change compared with the empirical moisture functions currently used in Earth system models.

---

[1] Institute of Surface-Earth System Science, Tianjin University, 300072 Tianjin, China. [2] Pacific Northwest National Laboratory-University of Maryland Joint Global Climate Change Research Institute, College Park, MD 20740, USA. [3] Pacific Northwest National Laboratory, Richland, WA 99354, USA. [4] State Key Laboratory of Environmental Geochemistry, Institute of Geochemistry, Chinese Academy of Sciences, 550081 Guiyang, China. [5] Guangdong Provincial Key Laboratory of Soil and Groundwater Pollution Control, School of Environmental Science and Engineering, Southern University of Science and Technology, 518055 Shenzhen, Guangdong, China. Correspondence and requests for materials should be addressed to C.L. (email: liucx@sustc.edu.cn)

Soil organic carbon (C) is the largest terrestrial C reservoir[1], and accurately predicting its decomposition rate in response to environmental factors is critical for projecting atmospheric carbon dioxide ($CO_2$) concentration and thereby climate change[2,3]. Next to temperature, moisture is the most important environmental factor controlling microbial heterotrophic respiration (HR)[2,4], which constitutes about half of the total $CO_2$ flux from soils[5]. Low moisture impedes HR rates by reducing solute transport through soils and may force microorganisms into dormancy under extremely dry conditions[6,7]. Conversely, high moisture restrains soil HR rates by suppressing oxygen ($O_2$) supply from the atmosphere[4]. The relationship between HR rates and moisture varies with soil types and characteristics[8–10], complicating the development of mechanistic models to predict the response of HR rates to moisture change, as well as introducing uncertainty into the projections of the feedback of soil C stocks to ongoing climate change[11,12].

Empirical moisture functions are commonly used in earth system models (ESMs) to account for the effects of moisture on C turnover rate in soils[13–15]. These functions are often statistically fitted using datasets from specific field sites, resulting in significant uncertainty when they are applied to other sites or expanded to regional and global scales[11]. Therefore, more general mechanistic models that incorporate underlying physicochemical and biological processes are needed to reduce the uncertainty in predicting $CO_2$ flux from soils[4,16]. Process-based models encapsulating effective diffusion of substrate and $O_2$, as well as microbial physiology and enzymatic kinetics have been developed to simulate and predict soil $CO_2$ flux[17–20]. Pore-scale models based on microbial behaviors have also been established to mechanistically examine organic C decomposition within soil aggregates[21], and were upscaled to simulate $CO_2$ flux at soil profile scales[22,23]. These mechanistic models described the HR–moisture relationship well for specific soils[17,20,22], but their applications to projecting the response of soil HR rates to moisture change for soils in general are strongly restrained,

mainly due to the lack of quantitative relationships between model parameters and soil properties[4,18,19].

Recently, Yan et al.[24] developed a microscale model to simulate the HR–moisture relationship. This model incorporated the primary physicochemical and biological processes controlling soil HR, and generated HR–moisture relationships in agreement with measurements for a heterogeneous soil core, elucidating how microscale heterogeneity of soil characteristics affects the relationships. However, such microscale modeling is computationally expensive and requires data on fine-scale soil properties, preventing its applications in large-scale modeling of soil C decomposition.

The objectives of this paper are to extend the work of Yan et al.[24] by developing a macroscopic moisture function, here termed $f_m$, that incorporates the underlying microscale processes controlling soil HR (Fig. 1), and to establish the quantitative relationship between parameters of $f_m$ and measurable soil properties to allow for the model's applications to different soils across spatial scales. First, $f_m$ was derived based on the primary physicochemical and biological processes controlling soil HR (see Methods section). Then, $f_m$ was tested using simulation results obtained by a microscale model modified from the previous Yan et al.'s model[24] (see Methods section). Furthermore, $f_m$ was evaluated and calibrated using published incubation data from a wide range of soil types. Lastly, $f_m$ was compared with laboratory and field measurements, as well as empirical models to assess its applicability and accuracy.

The moisture function derived in this study can be described by Eq. (1), hereafter referred to as $f_m$:

$$f_m = \begin{cases} \frac{K_\theta + \theta_{op}}{K_\theta + \theta} \left( \frac{\theta}{\theta_{op}} \right)^{1+an_s}, & \theta < \theta_{op} \\ \left( \frac{\phi - \theta}{\phi - \theta_{op}} \right)^b, & \theta \geq \theta_{op} \end{cases} \quad (1)$$

where $f_m$ is the relative HR rate (the value of which is normalized

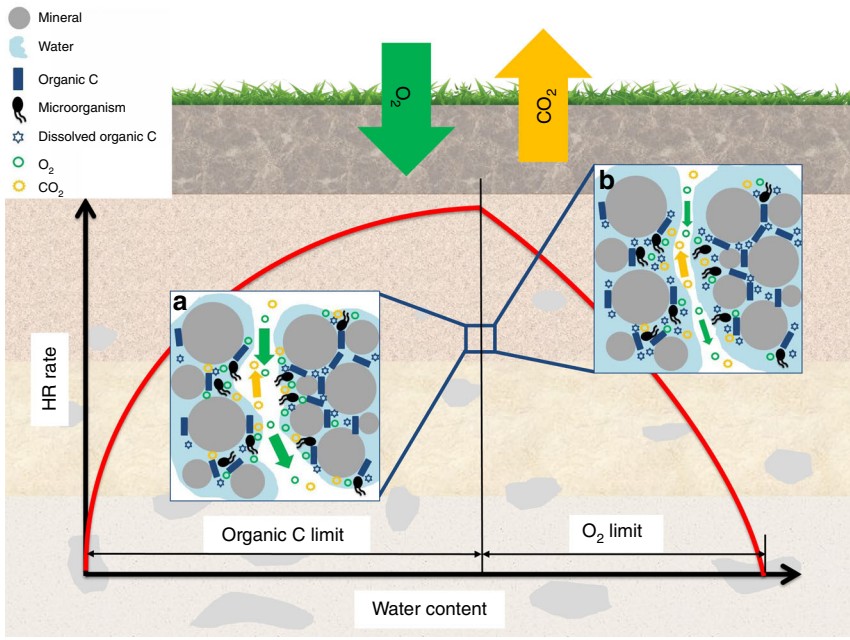

**Fig. 1** The macroscopic moisture function, $f_m$, and its links to microscale processes controlling heterotrophic respiration (HR) in soils. The red curve qualitatively describes the soil HR–moisture relationship, $f_m$, in soils. The inset conceptual figures depict the microscale processes controlling soil HR under dry conditions (**a**), in which HR rate is limited by the bioavailability of organic carbon (C), and under wet conditions (**b**), in which HR rate is limited by $O_2$ supply, respectively. The processes controlling the bioavailability of organic C and $O_2$ include the desorption of soil-adsorbed organic C, diffusion of dissolved organic C and $O_2$, and exchange between dissolved and gaseous $O_2$. See Methods section for the details of $f_m$ and the microscale processes

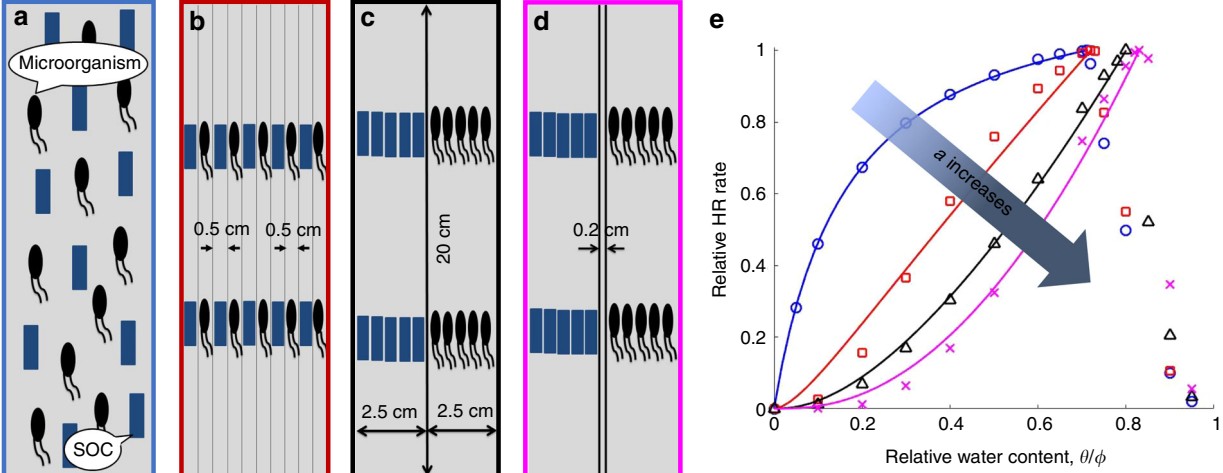

**Fig. 2** Effects of the collocation between soil-adsorbed organic carbon (SOC) and microorganisms on the SOC–microorganism collocation factor **a**. SOC and microorganisms are **a** uniformly allocated in the simulated soil core, **b** allocated in ten connected sections, **c** allocated in two connected sections, and **d** allocated in two separate sections. **e** The relative heterotrophic respiration (HR) rates change with the relative water content for the different allocations of SOC and microorganisms in **a–d**. The blue circles, red squares, black triangles, and pink crosses represent the simulated HR–moisture relationships obtained by the microscale model (see Methods section) for the different collocations in **a–d**, respectively. The simulated HR rates were calculated by averaging the $CO_2$ flux at the soil–atmosphere interface of the soil core. The solid curves represent the moisture function, $f_m$, with fitted $a$, $a = 0$, 0.40, 0.72, and 0.97, via the linear least-square regression

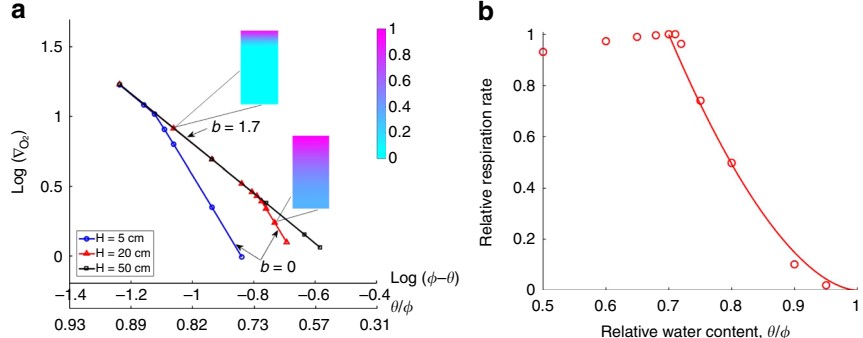

**Fig. 3** Effects of $O_2$ distribution on the $O_2$ supply restriction factor, $b$, in the simulated homogenous soil cores with different depths. **a** Change of $b$ with $O_2$ distribution, $b = \omega + 3.5$, where $\omega$ is the slope of $\log(\nabla_{O_2}) - \log(\phi - \theta)$ curves (see Eq. 8 in Methods section). The gradient of $O_2$ at the soil–atmosphere interface, $\nabla_{O_2}$, was calculated using the atmospheric $O_2$ concentration and the simulated $O_2$ concentration in the top numerical voxels of the soil cores. The inset plots show the distributions of the relative $O_2$ concentration with respect to the atmospheric $O_2$ concentration under different relative water contents ($\theta/\phi = 0.85$ and 0.68). SOC and microorganisms were completely collocated in the soil cores as in Fig. 2a. **b** Comparisons between the moisture function, $f_m$, and the simulated heterotrophic respiration (HR)–moisture relationship obtained by the microscale model (see Methods section) for the soil core with a depth of 20 cm. The red circles represent the simulation results, and the solid line represents $f_m$ with $b = 1.7$, which was fitted via the linear least-square regression

by the HR rate at $\theta_{op}$), $\theta$ is the water content, $\theta_{op}$ is the optimum water content at which the HR rate peaks, $K_\theta$ is a moisture constant reflecting the impact of water content on soil-adsorbed organic carbon (SOC) desorption (Eq. 2 in Methods section), $n_s$ is a saturation exponent reflecting the effects of pore connectivity on dissolved organic carbon (DOC) diffusion (Eq. 3 in Methods section), and $\phi$ is the soil porosity related to soil bulk density. As described in Fig. 1, soil HR is rate-limited by the bioavailability of organic C when $\theta < \theta_{op}$ and is rate-limited by $O_2$ supply when $\theta > \theta_{op}$. Two integrated parameters, $a$ and $b$, are introduced, mainly controlling the shape of $f_m$ when $\theta < \theta_{op}$ and $\theta > \theta_{op}$, respectively. Parameter $a$, the SOC–microorganism collocation factor, accounts for the effect of the collocation between SOC and microorganisms on HR rates (Eq. 5 in Methods section); parameter $b$, the $O_2$ supply restriction factor, accounts for the effect of $O_2$ supply on HR rates (Eq. 9 in Methods section). The function $f_m$ was tested and calibrated using simulation results and

published experimental data, and predicted the comparable HR–moisture relationships with laboratory and field measurements. Therefore, $f_m$ can potentially reduce the uncertainty in predicting the response of soil organic C stocks to climate change compared with the empirical moisture functions currently used in ESMs.

## Results

**Moisture function validation using microscale modeling.** Four different allocations of SOC and microorganisms (Fig. 2a–d) in a simulated soil core were used to test the hypothesis that the relationship between soil HR rates and water content can be described by the SOC–microorganism collocation factor $a$, $0 \le a \le 1$, when organic C is limiting (see Methods section). Figure 2e illustrates that the macroscopic moisture function, $f_m$, could capture the simulated HR–moisture relationships obtained by the

microscale model for the soil core with different SOC–microorganism collocations. The value of $a$ increased as the degree of collocation decreased, with $a = 0$ when SOC and microorganisms were uniformly distributed (Fig. 2a) and $a = 0.97$ when they were completely separated (Fig. 2d). These results were consistent for the simulated soil cores with different depths, porosity values, and organic C contents (Supplementary Fig. 1).

Simulated homogenous soil cores with different depths were used to test the hypothesis that the HR–moisture relationship can be described by the $O_2$ supply restriction factor, $b$, when $O_2$ is limiting (see Methods section). The simulation results illustrate that $b = 0$ when the bottom of the soil cores was aerobic and $b = 1.7$ when the bottom was anoxic (Fig. 3a), regardless of soil depth. Correspondingly, $f_m$ with $b = 1.7$ captured the simulated HR–moisture curve best in terms of the errors calculated by the least square (Fig. 3b). Importantly, the same results were found in the simulated homogenous soils with different porosity values, organic C contents, and saturation exponents (Supplementary Fig. 2). For simulated heterogeneous soil cores, the concentrations of $O_2$ at the bottom of soil cores shifted progressively from fully aerobic to anoxic as water content increased, resulting in a smooth change of $b$ from 0 to 1.7 (Supplementary Fig. 3a). Correspondingly, $f_m$ with $b = 1.4$ captured the simulated HR–moisture relationship best for the heterogeneous soil core (Supplementary Fig. 3b).

Simulated homogenous soil cores with different soil properties were further used to evaluate the analytical $\theta_{op}$ derived based on the assumption that water content is optimal when bioavailable DOC and $O_2$ are both limiting (see Methods section). Figure 4 shows that the analytical values of $\theta_{op}$ calculated using Eq. (11) in Methods section approximate the simulated ones obtained using the microscale model for the soil cores with a wide range of depths, porosity values, and SOC contents, especially when the values of $\theta_{op}$ are relatively small. The overprediction of the

analytical $\theta_{op}$ primarily emerged in the soil cores with shallow depths (see Supplementary Fig. 4).

**Moisture function calibration using incubation data.** Laboratory incubation data from different soil types were used to calibrate the moisture function $f_m$ (see Supplementary Data 1). Figure 5 shows the comparisons between the measured HR–moisture relationships and $f_m$ using fitted values of $a$ and $b$ for three soil types (see Supplementary Fig. 5 for comparisons of more soil types). The results show that $f_m$ generally described the measured HR–moisture relationships well for a wide range of soil types. Note that the value of $b$ is not available for soils whose HR rates were not measured under the condition of $\theta > \theta_{op}$, such as in Fig. 5b.

The SOC–microorganism collocation factor, $a$, is strongly related to soil clay content, $c_c$, and a linear relationship was derived based on the results from the different soil types (Fig. 6), $a = 2.8c_c - 0.046$. The findings indicate that the collocation between SOC and microorganisms decreases as clay content increases. By contrast, the $O_2$ supply restriction factor, $b$, shows no correlation with soil properties measured in the experiments. However, the observed values of $b$ varied mainly in a range between 0.5 and 1.0 (see Supplementary Data 1).

**Applications of the moisture function.** The applicability and accuracy of $f_m$ were assessed by comparing the predicted HR–moisture relationships with the measured ones in laboratory incubations[10] and field observations[25]. Figure 7 illustrates that $f_m$ generated comparable HR–moisture relationships with the measurements, especially for the laboratory incubations. The determination of $\theta_{op}$ is crucial for the accuracy of $f_m$ prediction. When using the measured $\theta_{op}$ (red coarse dash lines) instead of the analytical ones (blue coarse solid lines), $f_m$ better predicted both laboratory and field measurements. By simply assuming $\theta_{op}/\phi = 0.65$ (black coarse dotted lines), a value commonly observed in experiments of soil HR[4,18,26], the predicted HR–moisture relationships was not as good as using the analytical or measured $\theta_{op}$. However, Fig. 7 illustrates that it can be used as an approximation when neither the true nor the analytical $\theta_{op}$ is available.

As a final test, the predictions of $f_m$ were compared with empirical moisture functions commonly used in ESMs. Although in some cases using certain empirical functions, such as Myers[15] for the sandy loam, generated a comparable or even better HR–moisture relationship than $f_m$, none of the empirical functions performed well for both the sandy loam and loam (Fig. 7). In particular, when the value of the analytical $\theta_{op}$ was close to the true one, $f_m$ with parameter values estimated by soil properties generated a much better HR–moisture relationship than the empirical functions (Fig. 7a). Even if only soil bulk density and clay content were given, $f_m$ with the parameter values recommended by Table 1 in Methods section (black coarse dotted lines in Fig. 7) generally predicted the measured HR–moisture relationships better than most empirical functions.

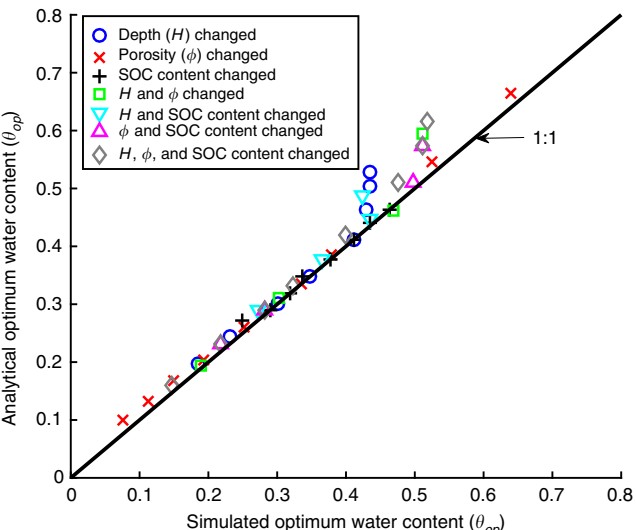

**Fig. 4** Evaluation of the analytical optimum water content $\theta_{op}$. Each symbol represents a comparison between the simulated and the analytical $\theta_{op}$ for a simulated homogenous soil core. The simulated $\theta_{op}$ was derived from the simulated heterotrophic respiration (HR)-moisture relationship obtained by the microscale model (see Methods section); the analytical $\theta_{op}$ was calculated using Eq. (11) in Methods section. The different symbols represent comparisons for different simulated soil cores with a wide range of depths $H$, $2.5 \leq H \leq 100$ cm, porosity values $\phi$, $0.2 \leq \phi \leq 0.8$, and SOC contents $C_{SOC}$, $0.005 \leq C_{SOC} \leq 0.2$ g g$^{-1}$. The base values were $H = 20$ cm, $\phi = 0.58$, and $C_{SOC} = 0.02$ g g$^{-1}$ in the simulations

**Discussion**

Establishing predictable relationships between soil HR rates and moisture is essential to evaluate the feedback of soil organic C stocks to ongoing climate change[27]. The current moisture functions that describe this relationship are mainly empirical and derived from single-site studies, introducing considerable uncertainty in projecting $CO_2$ flux from soils[15], e.g., the error caused by different empirical moisture functions were reported up to 4% of the global C stock by 2100[11]. This study develops a novel moisture function, $f_m$, by incorporating microscale processes that control soil HR, one that may improve the prediction of soil $CO_2$

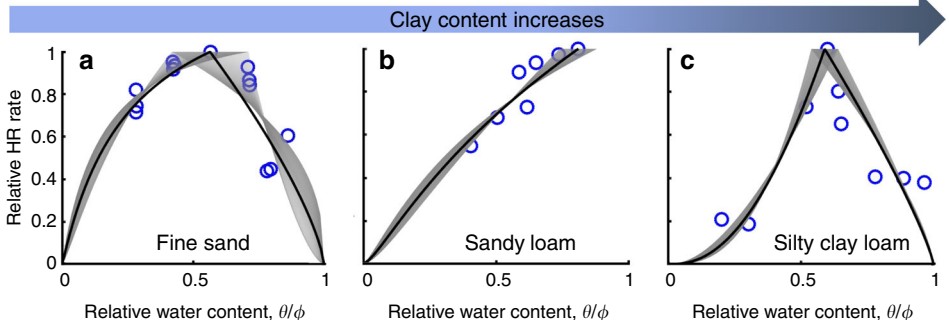

**Fig. 5** Calibration of moisture function, $f_m$, using incubation data. Blue circles are experimental data for different soil types: **a** fine sand[18], **b** sandy loam[25], and **c** silty clay loam[50]. Gray ranges represent the fitted $f_m$ with respect to different optimum water contents, $\theta_{op}$, whose values were allowed to vary in a range. The black lines represent the best-fit $f_m$ using the averaged values of $a$, $b$, and $\theta_{op}$ corresponding to the gray ranges (see Supplementary Data 1)

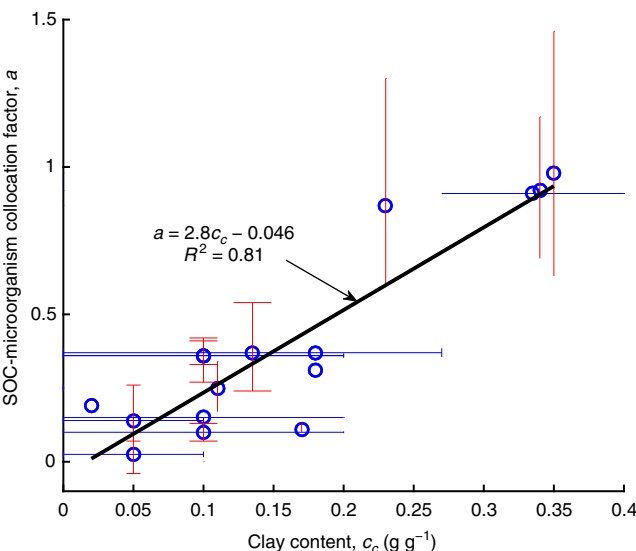

**Fig. 6** Relationship between the soil-adsorbed organic carbon (SOC)–microorganism collocation factor, $a$, and soil clay content, $c_c$. Blue circles represent the averaged values of $a$ and $c_c$ for the different soil types in Supplementary Data 1. The horizontal bars represent the ranges of $c_c$, which was not given in the literature and instead estimated according to the US Department of Agriculture textural triangle[76] (see Supplementary Data 1); the vertical bars represent the ranges of $a$, which corresponds to the ranges of the optimum water content (see Fig. 5, Supplementary Fig. 5). The black line was derived via the linear least-square regression[77]

flux in response to climate change. Different from previous mechanistic and process-based models that can only describe or predict the HR–moisture relationship for specific soil samples or field sites[17–23,28], $f_m$ is able to predict the relationship for different soils across spatial scales through establishing the quantitative relationships between the parameters of $f_m$ and measurable soil properties.

The function $f_m$ was developed by integrating theoretical derivation, numerical modeling, and experimental calibration. Theoretical derivation induced the mathematical expression of $f_m$; numerical modeling tested the assumptions used to derive $f_m$; and experimental calibration established the quantitative relationships between parameters of $f_m$ and measurable soil properties. In particular, microscale modeling enables us to assess hypotheses that are difficult or impossible to test using experimental approaches. For example, it is easy to examine how the

SOC–microorganism collocation quantitatively affects HR–moisture relationships in the microscale modeling, but is almost impossible in experiments due to the difficulty of controlling microbial distributions and activities[20,29]. Certainly, one needs to be careful in interpreting experimental observations using modeling results. For example, Fig. 2 illustrates that the SOC–microorganism collocation results in the different soil HR–moisture relationships, but other factors, such as microbial activity and nutrient availability, may be responsible for the relationships in natural soils[4,9]. In general, microscale models complement experimental tools, and provide a powerful means to study the effects of various biogeochemical processes on HR in natural soils[30], such as the effect of enzymes on SOC decomposition and associated HR rates[19]. Enzymes facilitate the breakdown of organic matter, and their distribution and transport are thus crucial for organic C turnover but difficult to measure in soils[7,19,31]. Simulation analysis using enzyme-related microscale models could help us to understand how enzymatic distribution, transport, and kinetics influence SOC decomposition.

The determination of parameter values is a key step in applying $f_m$. The SOC–microorganism collocation factor, $a$, linearly increased with soil clay content, reflecting the fact that the large surface areas of clay adsorb a large amount of organic C that cannot be accessed by microorganisms[3,32]. In addition, clay is crucial for aggregate formation that also potentially occludes organic C from microorganisms[33,34]. By contrast, the $O_2$ supply restriction factor, $b$, was not found mathematically relative to soil properties. This may be because the supply rate of $O_2$ at the soil-atmosphere interface is affected by not only the average soil properties but also their spatial distributions, and thus is difficult to describe or express by either single or multiple soil properties[35]. For example, soil porosity is an important indicator for $O_2$ availability[9], but pore connectivity is probably more crucial for aerobic respiration rates[10,36,37]. Similarly, both mass fraction and spatial distribution of soil texture influence $O_2$ distribution and diffusion, especially in well-structured loamy or clayey soils[38], in which pore size and water saturation are highly heterogenous[39,40]. Therefore, more experiments are needed to specify the value of $b$ and to establish its quantitative relationship with measurable soil properties, which should incorporate both general soil properties (bulk density, characteristic diameter of grain, clay content, specific surface area, etc.) and their spatial heterogeneity (pore and grain size distributions, aggregate distribution, etc.)[28,35,38,41]. In spite of the complexity, $b$ mainly varied in a restricted range between 0.5 and 1 (see Supplementary Data 1), illustrating certain degree of similarity in $O_2$ supply change in response to moisture variation for different soils. In addition, the value of the optimum water content, $\theta_{op}$, can be calculated using

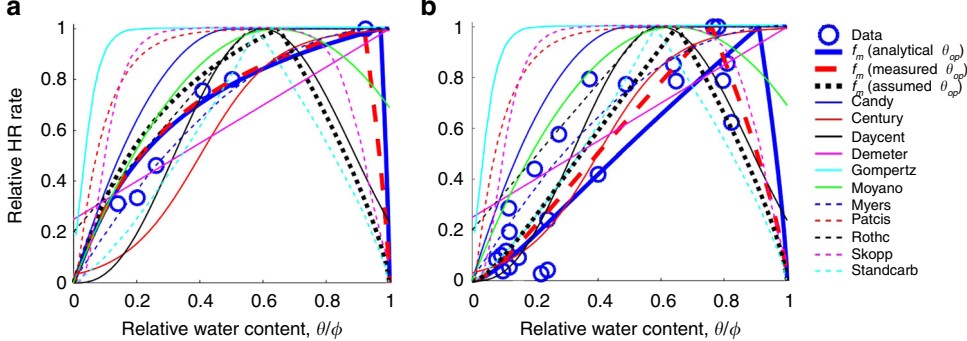

**Fig. 7** Applications of the moisture function, $f_m$, to laboratory and field observations. Comparisons of $f_m$ with measured heterotrophic respiration (HR)-moisture relationships for **a** sandy loam in laboratory incubations[10] and **b** loam in field observations[25]. The coarse blue solid lines are the predicted HR-moisture relationship using $f_m$ with $a = 2.8c_c - 0.046$, $b = 0.75$, and $\theta_{op}$ calculated by Eq. (11) in Methods section. The coarse red dash and black dotted lines represent $f_m$ using the same $a$ and $b$ values as in the coarse blue solid lines but measured $\theta_{op}$ and assumed $\theta_{op}$, $\theta_{op} = 0.65\phi$, respectively. Other lines represent widespread empirical moisture functions commonly used in Earth system models (see Supplementary Table 2)

soil properties (Eq. 11 in Methods section). When soil properties required to determine the parameter values of $f_m$ are not available, recommended values are provided for practical applications (see Table 1 in Methods section). These recommended values significantly simplify the application of $f_m$, and produced comparable predictions with measurements (Fig. 7).

The function $f_m$ provides a simple way to assess the effects of soil properties on soil HR-moisture relationships. For example, when clay content, $c_c$, increases, HR rates apparently decrease under relatively dry conditions and the value of $\theta_{op}$ increases (Supplementary Fig. 6a). Such results have been observed in laboratory experiments in which different amounts of clay were added to examine their impact on soil HR rates[25,39,42], and are also consistent with the analysis from a regression model based on incubation data[9]. Contrarily, when soil depth, $H$, or organic C content, $C_{SOC}$, increases, HR rates increase under relatively dry conditions and the value of $\theta_{op}$ decreases (Supplementary Fig. 6c, e). In particular, the soil HR-moisture relationship is not sensitive to the change of organic C in deep or C-rich soils, implying a relatively weak sensitivity of deep and rich organic C decomposition to moisture variation. This finding provides insights for projecting soil $CO_2$ flux in response to climate change, given that a large amount of organic C is stocked in deep soils below 1 m and in peatland with high organic C content[2,43,44]. Moreover, the HR-moisture relationship is found to be insensitive to the diffusion-related exponents (Supplementary Fig. 6i–l), consistent with experimental measurements from different textured soils[45]. The usage of $n_s = 2$ is thus a safe approximation for the application of $f_m$ to different soil types (Table 1 in Methods section). By contrast, the HR-moisture relationship is relatively sensitive to the organo-mineral-related parameters (Supplementary Fig. 6f–h), whose values varied up to two orders of magnitude for different organic C species and mineral components[46,47]. Therefore, more experiments are required to specify the values of these organo-mineral-related parameters, especially $K_\theta$, to refine and test the predictions of $f_m$.

Like any model, $f_m$ represents a trade-off between the convenience of model development and application, and the complexity of soil HR, and it is important to note that the development of $f_m$ adopted assumptions and simplifications that may invalidate its application in some natural soils. For example, we implicitly assumed that the concentration and gradient of DOC do not affect the DOC release rate (Eq. 2 in Methods section), which is valid only when the concentration of DOC is low. If the SOC is highly concentrated or is distant from microorganisms, such as inside aggregates, fast desorption or slow

diffusion of DOC may result in its accumulation around the SOC and thus reduce the DOC release rate[19,33,48]. Moreover, we quantified the parameter $a$ using only clay content and constrained its values to between 0 and 1 by assuming $a = 0$ and 1 under extremely low and high clay content, respectively. However, silt and sand also affect the SOC–microorganism collocation[36,49], and thus influence soil HR-moisture relationships[37]. The value of $a$ could be below 0 at very low clay contents, or above 1 at very high clay contents[28,50]. In addition, we neglected the effect of water and air percolation thresholds on DOC and $O_2$ diffusion (Eqs. (5) and (9) in Methods section), given than the percolation thresholds primarily affect the HR-moisture relationship under extreme water saturation and mainly reduce the absolute rather than the relative HR rates[24].

Similarly, $f_m$ is based on the competitive diffusion between organic C and $O_2$, which may disable $f_m$ to capture the soil HR-moisture relationship for situations in which other mechanisms are of paramount importance[3,4]. For example, microorganisms may die or shift to dormancy under extremely dry conditions[7], situations in which microbial physiology rather than C diffusion determines respiration rates[51]. When dry soils rewet, the respiration rate may rapidly increase for a brief period partly due to the nutrients suddenly available for microbial activity[52,53], a phenomenon known as the Birch effect[4,54]. Furthermore, alternative electron acceptors such as $NO_3^-$ and $SO_4^-$ can interact with organic C and produce $CO_2$ under wet conditions[50,54,55]. Even under dry conditions, the heterogeneous distribution of water in soils may form anoxic microsites where anaerobic respiration occurs[56]. These missing mechanisms together with the aforementioned assumptions and simplifications should contribute to the failure of $f_m$ in capturing some features of measured HR-moisture relationships, such as the plateau around the maximum HR rate observed in experiments[52,57,58]. This plateau is mostly expected to emerge in soils maintaining strong anaerobic HR, which counteracts the negative effect of $O_2$ depletion[59,60]. Therefore, we argue that $f_m$ is most suitable for soils whose moisture levels are not extreme, in which the diffusion limits of organic C and $O_2$ control HR rates[24,53].

The performance of $f_m$ was assessed by comparison with measured HR-moisture relationships from both laboratory experiments and field observations (Fig. 7). $f_m$ generally agreed well with the measured relationships especially with the laboratory data. The relatively large deviation between $f_m$ and the field data could be attributed to factors that influence the HR-moisture relationship in natural environments. For example, the bioavailability of organic C and $O_2$ in the field is determined

by advection and dispersion rather than diffusion during hydrological disruptions, such as precipitation and drainage, in which massive organic matter leaches into deep soils[43,61]. Even without such hydrological disruption, water movement caused by plant transpiration may dominate the transport of organic C in regions around the rhizosphere[62]. Moreover, natural soils often feature aggregates and macro-pores as well as fractures, complicating the transport and distribution of $O_2$[63]. In particular, aggregates constrain $O_2$ diffusion and may reduce $O_2$ bioavailability for microbial HR[21], while macro-pores and fractures facilitate $O_2$ supply by delivering $O_2$ into deep soils[63]. By contrast, laboratory incubation experiments often use sieved soils that destroy macro-pores and fractures, in which soils the $O_2$ supply rate tends to reduce more fast than in natural soils as water content increases[10,50]. In addition, temperature always affects soil HR rates by modifying microbial activity and solute diffusion[2,64]. The changes in temperature and in the availability of organic C and $O_2$ make the HR–moisture relationship vary dynamically in the field. Despite these potential factors, $f_m$ performed better than most commonly used empirical moisture functions (Fig. 7b), demonstrating its ability and robustness.

In summary, this study develops a novel moisture function, $f_m$, by incorporating the microscale processes that control soil HR, and establishes quantitative relationships between the parameters of $f_m$ and measurable soil properties. The feasible application of $f_m$ enables it to predict the HR–moisture relationships for different soils across spatial scales, potentially reducing the uncertainty of modeled C cycles in ESMs by improving on their current empirical moisture functions. The function $f_m$ demonstrated its applicability in predicting the HR–moisture relationships from laboratory experiments and field observations, although some mechanisms of soil HR are neglected in $f_m$. These mechanisms can be taken into account in future investigation using a similar strategy as in this study, i.e., first incorporated in a microscale context and then encapsulated into a macroscopic model.

## Methods

**Moisture function**. Function development: The moisture function, $f_m$, was developed based on the primary physicochemical and biological processes controlling HR in soils. Organic carbon (C) is assumed to initially adsorb onto soil mineral surfaces and is consumed by microorganisms after two steps: the SOC converts to DOC after desorption, and the DOC is diffused to regions where microorganisms inhabit.

The flux of DOC released from SOC can be estimated by[24]

$$F_{DOC}^{total} = \frac{\theta}{K_\theta + \theta} \alpha m_{SOC} \quad (2)$$

where $\theta$ is water content [$m^3\,m^{-3}$], $K_\theta$ is a moisture constant reflecting the impact of moisture content on C desorption [$m^3\,m^{-3}$][24], $\alpha$ is the mass transfer coefficient between SOC and DOC [$s^{-1}$][46], and $m_{SOC}$ is organic C content per unit area of soils [$kg\,m^{-2}$]. The released DOC is biologically degraded after diffusing into regions containing microorganisms, thus the turnover rate of SOC is related to the degree of collocation between SOC and microorganisms.

In soils where SOC and microorganisms are completely separated, the flux of bioavailable DOC for HR can be described by[45]

$$F_{DOC} = F_{DOC}^{total} \phi^{(m_s - n_s)} \theta^{n_s} \quad (3)$$

where $\phi$ is soil porosity [−], $m_s$ and $n_s$ are cementation and saturation exponents [−], accounting for the effects of pore structure and water connectivity on DOC diffusion[45].

In soils where SOC and microorganisms are completely collocated, the released DOC can be degraded locally without diffusion. Therefore, the flux of bioavailable DOC for HR is the same as the flux of total available DOC

$$F_{DOC} = F_{DOC}^{total} \quad (4)$$

For most soils, microorganisms are partly separated from SOC: released DOC is degraded either locally or after diffusion. We introduce a parameter $a$, the SOC–microorganism collocation factor, to represent the degree of collocation between SOC and microorganisms. Consequently, the flux of bioavailable DOC for

soil HR can be described by

$$F_{DOC} = F_{DOC}^{total} \phi^{a(m_s - n_s)} \theta^{an_s} \quad (5)$$

where $a$ increases as the degree of collocation between SOC and microorganisms decreases. Given that $a = 0$ when SOC and microorganisms are completely collocated (Eq. 4) and $a = 1$ when they are completely separated (Eq. 3), $0 < a < 1$ is presumed when they are partly collocated. Therefore, Eq. 5 with $0 \le a \le 1$ uniformly describes the relationship between HR rates and water content for soils with full degrees of collocation between SOC and microorganisms.

When soil HR is limited by organic C bioavailability, the HR rate is determined by the flux of bioavailable DOC and its response to water content should be the same as for the flux of bioavailable DOC. Therefore, we hypothesize that, when organic C is limiting, the relationship between soil HR rates and water content can be described by the SOC–microorganism collocation, which is represented by $a$ as in Eq. (5).

Soil HR becomes $O_2$ limited when water content is above the optimal value, $\theta > \theta_{op}$. Considered that $O_2$ diffusion through liquid can be ignored compared with that through air[65], the supply rate of $O_2$ from the atmosphere to soils can be estimated using the gaseous $O_2$ diffusion at the soil–atmosphere interface,

$$F_{O_2} = D_{GO} \nabla_{O_2} (\phi - \theta) \quad (6)$$

where $D_{GO}$ is the effective diffusion coefficient of gaseous $O_2$ at the soil–atmosphere interface [$m^2\,s^{-1}$], and can be estimated by[66]

$$D_{GO} = \phi^{m_g - n_g} (\phi - \theta)^{n_g} D_{GO,0} \quad (7)$$

where $m_g$ and $n_g$ are cementation and saturation exponents accounting for the effects of pore structure and air connectivity on $O_2$ diffusion in soils[45], respectively, and $D_{GO,0}$ is the diffusion coefficient of $O_2$ in pure air [$m^2\,s^{-1}$]. $\nabla_{O2}$ is the gradient of gaseous $O_2$ concentrations between the top soil surface and the atmosphere [$g\,l^{-1}\,m^{-1}$], and can be expressed by[18]

$$\nabla_{O2} = k_{GO} (\phi - \theta)^\omega \quad (8)$$

where $k_{GO}$ is a coefficient representing the degree of oxygen depletion in soils, and $\omega$ reflects the impact of soil pore connectivity on $O_2$ transport. Substituting Eqs. (7) and (8) into Eq. (6), we have

$$F_{O2} = k_{GO} \phi^{m_g - n_g} (\phi - \theta)^b D_{GO,0} \quad (9)$$

where $b$, $b = 1 + n_g + \omega$, is a parameter reflecting the effects of soil characteristics on $O_2$ supply at the soil–atmosphere interface, called the $O_2$ supply restriction factor.

The supplied $O_2$ diffuses into soils and enters water to form dissolved oxygen (DO), which is eventually consumed by microorganisms. Regardless of $O_2$ delivery from plant roots, the flux of bioavailable DO for soil HR should be the same as the flux of $O_2$ supply from the atmosphere

$$F_{DO} = F_{O_2} \quad (10)$$

When soil HR is limited by $O_2$ bioavailability, its rate response to water content should be the same as for the flux of bioavailable DO. Therefore, we hypothesize that, when $O_2$ is limiting, the relationship between soil HR rates and water content can be described by the $O_2$ supply restriction factor, $b$, as shown in Eq. (9).

Theoretically, the soil HR rate maximizes when bioavailable DOC and DO are both limiting[18], i.e., the supplied DO is stoichiometrically enough to react with the bioavailable DOC, $F_{DO} = v_{DO} F_{DOC}$, where $v_{DO}$ is the stoichiometric coefficient of DO with respect to DOC [$g\,g^{-1}$]. Correspondingly, the water content was regarded as optimum water content, $\theta_{op}$, that can be calculated by

$$v_{DO} \frac{\theta_{op}}{K_\theta + \theta_{op}} \alpha m_{SOC} \phi^{a(m_s - n_s)} \theta_{op}^{an_s} = k_{GO} \phi^{m_g - n_g} \left( \phi - \theta_{op} \right)^b D_{GO,0} \quad (11)$$

Considered only water content related terms and normalized to the maximum HR rate, the process-based moisture function, $f_m$, can be expressed by

$$f_m = \begin{cases} \frac{K_\theta + \theta_{op}}{K_\theta + \theta} \left( \frac{\theta}{\theta_{op}} \right)^{1 + an_s}, & \theta < \theta_{op} \\ \left( \frac{\phi - \theta}{\phi - \theta_{op}} \right)^b, & \theta \ge \theta_{op} \end{cases} \quad (12)$$

Function parameterization and evaluation: Parameter and initial values used in the simulations are presented in Supplementary Table 1. For the evaluation of analytical $\theta_{op}$ (Fig. 4), $a = 1$ and $b = 1.7$ were used to calculate the values of analytical $\theta_{op}$ because homogeneous soils were utilized, and $m_{SOC}$ was calculated using $\rho_s(1 - \phi) HC_{SOC}$ where $\rho_s$ is the density of soil mineral[39]. $k_{GO}$ was estimated

**Table 1 Determinations of parameter values in the application of $f_m$**

| Parameters | Descriptions | Value estimated by soil properties | Value recommended if not available | sources and notes |
|---|---|---|---|---|
| $a$ | SOC–microorganism collocation factor | Can be estimated by clay content ($c_c$) $$a = \begin{cases} 0, c_c \leq 0.016 \\ 2.8c_c - 0.046, 0.016 < c_c \leq 0.37 \\ 1, c_c > 0.37 \end{cases}$$ | — | Fig. 6 |
| $b$ | $O_2$ supply restriction factor | Depend on $O_2$ supply $0 \leq b \leq 1.7$ | 0.75 | Supplementary Data 1 |
| $\theta_{op}$ | Optimum water content | Can be calculated implicitly by soil properties $$\nu_{DO}\frac{\theta_{op}}{K_\theta + \theta_{op}}\alpha m_{SOC}\phi^{a(m_s - n_s)}\theta_{op}^{an_s} = k_{GO}\phi^{m_g - n_g}(\phi - \theta_{op})^b D_{GO,0}$$ | 0.65$\phi$ | 4, 18 |
| $\phi$ | Soil porosity | Can be estimated by soil bulk density ($\rho_b$) and mineral density ($\rho_s$) $$\phi = 1 - \frac{\rho_b}{\rho_s}$$ | — | 67 |
| $n_s$ | Saturation exponent | Depend on soil structure and texture | 2 | 45 |
| $K_\theta$ | Moisture constant | Depend on organo-mineral associations | 0.1 | 24 |

by the intercepts of $\log(\nabla_{O2}) - \log(\phi - \theta)$ curves with $y$-coordinate, whose value equals $\log(k_{GO}) + \omega \log(\phi - \theta)$ (Fig. 3a). The simulation results showed that $k_{GO}$ primarily changed with SOC content and its value could be estimated using $k_{GO} = 0.7465C_{SOC}^{0.512}$ (Fig. 3a, Supplementary Fig. 2).

For the applications of $f_m$ to laboratory and field observations (Fig. 7), the values of $\alpha$ and $\nu_{DO}$ were not given in the literature and were estimated using Eq. (11) in Methods section, in which $a$, $b$, and $\theta_{op}$ were fitted using the measured data. Note $b = 0.75$ was used for the sandy loam in the laboratory incubation[10], where the measured HR rates were available only for $\theta < \theta_{op}$. Consequently, $\alpha \times \nu_{DO} = 3.56 \times 10^{-8} \text{ s}^{-1}$ for the sandy loam[10] and $2.9 \times 10^{-7} \text{ s}^{-1}$ for the loam[25].

Function application: The application of $f_m$ requires to determine six parameters. The SOC–microorganism collocation factor, $a$, can be estimated using clay content $c_c$, $a = 2.8c_c - 0.046$. For soils with low clay content ($c_c < 0.016 \text{ g g}^{-1}$), we assume $a = 0$; for soils with high clay content ($c_c > 0.37 \text{ g g}^{-1}$), we assume $a = 1$. The $O_2$ supply restriction factor, $b$, is assumed as constant, $b = 0.75$, in practical applications. The optimum water content, $\theta_{op}$, can be calculated implicitly using Eq. (11) in Methods section, and its value depends on soil properties, as well as the values of $a$ and $b$. If these properties are not available, we assume $\theta_{op}/\phi = 0.65$, a value widely observed in laboratory and fields[4,18,26]. The soil porosity, $\phi$, can be estimated using bulk density $\rho_b$[67], $\phi = 1 - \frac{\rho_b}{\rho_s}$. The saturation exponent, $n_s$, is relatively invariable, and can be assumed to be constant[45], $n_s = 2$. The moisture constant, $K_\theta$, depends on organo-mineral associations, and its value can be assumed to be constant, $K_\theta = 0.1$, when unavailable[24]. The determination of parameter values in the application of $f_m$ were summarized in Table 1.

**Microscale model.** Microscale processes: The original microscale model developed by Yan et al.[24] was simplified by neglecting the effects of water and air percolation to test the moisture function, $f_m$, developed in this study. Important processes that affect soil HR rates are considered in the microscale model. These processes include organic C partition between adsorbed and dissolved phases, $O_2$ and $CO_2$ diffusion and partition in gas and liquid phases, and microbial metabolism of DOC and DO. The transformation of SOC to DOC was described using a first-order kinetic model[68]. Microbial metabolism was described using the dual Monod model with respect to DOC and DO. The biogenic $CO_2$ formed various dissolved inorganic carbon species, which were assumed to be in local equilibrium with gas phase $CO_2$ following the Henry's law[69]. The gas phase $CO_2$ was allowed to release into the atmosphere through the top surface of soils. The dissolved and gaseous $O_2$ were also assumed to be at local equilibrium following the Henry's law[69], and were supplied through diffusion from the top soils where they were in equilibrium with the atmospheric $O_2$. With these treatments, the controlling processes of HR in soils could be described by[24]:

$$\frac{\partial C_{DOC}}{\partial t} - \nabla \cdot (D_{DOC}\nabla C_{DOC}) = \frac{\rho_s(1-\phi)}{\theta}\frac{\theta}{K_\theta + \theta}\alpha(C_{SOC} - K_c C_{DOC}) - k_{DOC}C_B\frac{C_{DOC}}{C_{DOC} + K_{DOC}}\frac{C_{DO}}{C_{DO} + K_{DO}}$$
(13)

$$\frac{\partial C_{SOC}}{\partial t} = -\frac{\theta}{K_\theta + \theta}\alpha(C_{soc} - K_c C_{DOC})$$
(14)

$$\frac{\partial C_B}{\partial t} = Y k_{DOC} C_B \frac{C_{DOC}}{C_{DOC} + K_{DOC}}\frac{C_{DO}}{C_{DO} + K_{DO}} - k_B C_B$$
(15)

$$\frac{\partial(\theta C_{DO} + (\phi - \theta)C_{GO})}{\partial t} - \theta\nabla \cdot (D_{DO}\nabla C_{DO}) - (\phi - \theta)\nabla \cdot (D_{GO}\nabla C_{GO}) = -\theta\nu_{DO}k_{DOC}C_B\frac{C_{DOC}}{C_{DOC} + K_{DOC}}\frac{C_{DO}}{C_{DO} + K_{DO}}$$
(16)

$$\frac{\partial(\theta C_{DIC} + (\phi - \theta)C_{GIC})}{\partial t} - \theta\nabla \cdot (D_{DIC}\nabla C_{DIC}) - (\phi - \theta)\nabla \cdot (D_{GIC}\nabla C_{GIC}) = \theta\nu_{DIC}k_{DOC}C_B\frac{C_{DOC}}{C_{DOC} + K_{DOC}}\frac{C_{DO}}{C_{DO} + K_{DO}}$$
(17)

$$C_{DO} = K_{h,o}C_{GO}$$
(18)

$$C_{DIC} = K_{pH}K_{h,c}C_{GIC}$$
(19)

where $C_{DOC}$ is the DOC concentration [g l$^{-1}$], $C_{SOC}$ is the SOC content [g g$^{-1}$], $C_B$ is the concentration of microorganisms [g l$^{-1}$], $C_{DO}$ is the DO concentration [g l$^{-1}$], $C_{GO}$ is the concentration of gaseous $O_2$ [g l$^{-1}$], $C_{DIC}$ is the concentration of dissolved inorganic carbon (DIC) [g l$^{-1}$], $C_{GIC}$ is the concentration of gaseous $CO_2$ [g l$^{-1}$], $\phi$ is soil porosity whose values may be different for each numerical voxel in microscale simulations [–], $\theta$ is water content which was assumed the same for all numerical voxels [m$^3$ m$^{-3}$], $K_c$ is the adsorption/desorption equilibrium constant of DOC [l g$^{-1}$], $Y$ is the yield coefficient of microbial biomass [g g$^{-1}$], $k_B$ is the first-order decay coefficient of microorganisms [1 s$^{-1}$], $k_{DOC}$ is the maximum rate of DOC metabolism [g g$^{-1}$ s$^{-1}$], $K_{DOC}$ is the half-rate coefficient with respect to DOC [g l$^{-1}$], $K_{DO}$ is the half-rate coefficient with respect to DO [g l$^{-1}$], $\nu_{DIC}$ is the stoichiometric coefficient of DIC with respect to DOC [g g$^{-1}$], $D_{DOC}$ is the effective diffusion coefficient of DOC [m$^2$ s$^{-1}$], $D_{DO}$ is the effective diffusion coefficient of DO [m$^2$ s$^{-1}$], $D_{GO}$ is the effective diffusion coefficient of gaseous $O_2$ [m$^2$ s$^{-1}$], $D_{DIC}$ is the effective diffusion coefficient of DIC [m$^2$ s$^{-1}$], $D_{GIC}$ is the effective diffusion coefficient of gaseous $CO_2$ [m$^2$ s$^{-1}$], $K_{h,o}$ is the Henry constant for $O_2$ [–], $K_{h,c}$ is the Henry constant for $CO_2$ [–], and $K_{pH}$ is a coefficient related to equilibrium reactions of DIC species and pH value:

$$K_{pH} = 1 + \frac{K_{a1}}{[H^+]} + \frac{K_{a1}K_{a2}}{[H^+]^2}$$
(20)

where $K_{a1}$ and $K_{a2}$ are two equilibrium carbonic acid speciation constants [mole m$^{-3}$][70].

The diffusivity of dissolved and gaseous species in soils depend on the moisture saturation degree, as well as pore water and pore air connectivity[35]. In this study, this dependency is described using the following equations[71,72]

$$\frac{D_D}{D_{D,0}} = \phi^{m_s - n_s}\theta^{n_s}$$
(21)

$$\frac{D_G}{D_{G,0}} = \phi^{m_g - n_g}(\phi - \theta)^{n_g}$$
(22)

where $D_D$ and $D_G$ are the effective diffusion coefficients of dissolved species [m$^2$ s$^{-1}$] (e.g., $D_{DOC}$, $D_{DO}$, and $D_{DIC}$ in Eqs. (13), (16), and (17)) and gaseous species [m$^2$ s$^{-1}$] (e.g., $D_{GO}$ and $D_{GIC}$ in Eqs. (16) and (17)), respectively, $D_{D,0}$ and $D_{G,0}$ are the corresponding diffusion coefficients in pure water and air [m$^2$ s$^{-1}$], respectively, $m_s$ and $m_g$ are cementation exponents [–], $n_s$ and $n_g$ are saturation exponents [–]. $m_s$, $n_s$ and $m_g$, $n_g$ are parameters considering the effects of tortuosity and pore connectivity on diffusion of dissolved and gaseous species, respectively[45,73].

Simulation and parameterization procedures: A previously developed code was used to solve the governing equations (Eqs. 13–17), and the solving process was reported in the previous study[74]. A spatial spacing of 20 μm was applied to discretize the soil cores used in the simulations. The initial SOC concentration was assumed to be proportional to solid mass fraction in each numerical voxel[24]. No DOC was assumed to exist initially in the simulated soil cores. A measured microbial concentration was used as the initial biomass concentration[75]. The initial

concentrations of the gaseous $O_2$ and $CO_2$ in soils were assumed to be in equilibrium with the atmospheric concentrations under 1 atm and 25 °C. The initial concentrations of dissolved $O_2$ and $CO_2$ were assumed in equilibrium with gaseous $O_2$ and $CO_2$ following the Henry's law. pH value was assumed to be constant (pH = 6.8). Since the top surfaces of the simulated soil cores were assigned to connect to the atmosphere, the concentrations of $O_2$ and $CO_2$ were fixed on the top boundary. No flux boundary condition was applied to the walls. The initial and parameter values are presented in Supplementary Table 1. SOC content and microorganism concentration were fixed during the simulations to produce steady-state $CO_2$ flux.

Model calibration: Simulated and measured results were first compared to evaluate the effectiveness of the microscale model in simulating soil HR rates as a function of $\theta$ (Supplementary Fig. 7). The measured data in Supplementary Fig. 7 are from literature where soil samples were incubated in canning jars under different water saturation conditions for 24 days[26]. This literature reported the relationship between HR rates and $\theta$ for natural soils with different SOC contents and bulk densities. The measured HR rates under different moisture saturation degrees for all natural soils were used to validate the microscale model to reduce the uncertainty caused by the different SOC contents and bulk densities. Since the experiment did not provide spatial structures of the soil cores necessary for microscale simulations, we created a homogeneous soil core to mimic the soil cores used in the experiments. The simulated homogenous soil core had the same size as the cores used in the experiment; the values of porosity, $\phi$, and SOC content, $C_{SOC}$, were the averaged ones over all the natural soils used in the experiments, $\phi = 0.58$ and $C_{SOC} = 0.02$ g/g. The simulated HR–$\theta$ relationship agreed well with the measured one (Supplementary Fig. 7), indicating that the microscale model can capture the HR rates observed in the core-scale experiment.

**Data availability**. The measured data used to evaluate and calibrate the moisture function, $f_m$, was summarized in Supplementary Data 1. The codes used to calculate the optimum water content, $\theta_{op}$, and to determine $f_m$ are provided in Supplementary Software 1–2. All the data and codes have been deposited on Figshare—DOI: 10.6084/m9.figshare.6337574. The codes of the microscale model used in this study will be available from the authors upon request (liucx@sustc.edu.cn, yanzf17@tju.edu.cn), and will be deposited in the same folder on Figshare in the near future.

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

## Acknowledgements

This research was supported by the US Department of Energy (DOE) Office of Science, Biological and Environmental Research (BER) Division through the Terrestrial Ecosystem Science (TES) program, and the National Key R&D Program of China (2016YFA0601002) . PNNL is operated by Battelle Memorial Institute under subcontract DE-AC06-76RLO 1830. K. Todd-Brown is grateful for support given by the Linus Pauling Distinguished Postdoctoral Fellowship, a Laboratory Directed Research program at PNNL. C. X. Liu would like to acknowledge the support from the National Natural Science Foundation of China (Nos. 41572228 and 41521001), by the Program for Guangdong Introducing Innovative and Entrepreneurial Teams (2017ZT07Z479), and additional support from Southern University of Science and Technology (G01296001).

## Author contributions

This study was conceived by C. X. L., who also guided the model development and manuscript writing. Z. F. Y. developed the model, performed the numerical simulations, and drafted the manuscript. K. T. B. helped to develop the model and the writing. B. B. -L., V. L. B., and C. Q. L. supported the research and helped with revision of the manuscript. S. L. guided to calibrate and test the model using experimental data.

## Additional information

**Competing interests:** The authors declare no competing interests.

