## [Peer Review File · Nature Communications]

Reviewers' comments:

Reviewer #1 (Remarks to the Author):

Review

The paper by Yan et al. presents an interesting study describing a new function relating soil heterotrophic respiration with moisture content. The topic is of clear importance for modeling the carbon cycle in terrestrial ecosystems and contributions such as this have the potential to improve the predictive accuracy of general land surface models.

The dependency of the response of microbial activity to soil water content is indeed something that needs to be improved in soil C models. This study uses both model and data approaches to try to derive a generally applicable function that captures such dependencies. It applies some novel formulations, in particular by combining calculations of diffusivity with a parameter representing microbial-SOC collocation or with an oxygen supply restriction factor, depending on the case. The main message in the paper is that this new function is an improvement over previous functions as it correctly reproduces data obtained either from pore-scale simulations or lab observations.

And here I get to the paper's main weak point. This is a study that is relatively heavy on equations, which are needed to understand what was done. A careful step by step description is required to go from the general problem to the solution. It is a complex topic that requires close examination. However, the paper was written in the letter format trying to very succinctly describe the work. I strongly believe that, while a short message can be delivered for this study (a new function was developed that improves over previous ones), the main part should consist of a clear explanation of the analysis itself and its implications, something that is incompletely described in the main text. This study should be about carefully explaining how and why the development was done and tested, something that requires a longer article format. The way it is now, the reader needs to shift between main text, methods and supplementary material in order to understand what is being described. The main text is not self-explanatory and does not read easily because of the above.

There are also a number of points that are not clear and need addressing.

In summary, I believe the study is very publishable and can become a significant contribution to the topic. However, the paper needs to be improved before publication, and will largely benefit from changing the format, probably leading to more citations afterwards.

Specific remarks:

It was not clear to me how the function was derived from the pore-scale model. It may be based on some of the same equations used for the pore scale model, such as the desorption or diffusion flux. But these are general equation, unless they were parameterized with the pore scale mode. What seems the case is that the function was validated against the pore scale model.

In several parts it was not clear if the pore-model results or the measured data was being referred to.

What happens to the part of $F_{total}[DOC]$ that does not diffuse to the microbes? This will add to a pool that should affect the flux.

Oxygen flux: why is the oxygen gradient made to depend on SOC and not on the actual $[O_2]$ in the soil? What happens with deeper layers?

Not clear what 'bottom of soils' refers to. That is transfers from aerobic to anaerobic is also not clear.

Why is it important to calculate θ_{op} ? It would be enough to express the function as the minimum of the substrate supply and oxygen supply functions.

'DOC and DO are both completely degraded'. This phrase is not clear, especially since DO has not been defined anywhere.

The final function eq(11) depends on a and b. However, a

Should distinguish between diffusion and diffusivity. Apart from the effects of texture on diffusivity introduced in eq(4), the concentration gradient and diffusive distance should influence the total flux. Even if these are implicitly dealt with in the function, they should be discussed. In this same context, recent previous works that deal with similar topics should be cited (Manzoni et al. 2016, Modeling coupled enzymatic and solute transport controls on decomposition in drying soils).

Given that a dependency with clay content is found, it would be appropriate to compare it to previous related finding, namely Moyano et al. 2012, already cited in the text.

Data used to fit the relationship between 'a' and clay seems the same as used for validating the function. Independent data should be used. Related to this: the 'wide range' of published data used to test the function seems to be only 3 studies. Not a wide range of published data. Maybe of clay content. but it is not clear in the paper if the function actually performed well against independent observations over a wide range of soil types. In addition, it seems like the function cannot simulate a plateau of optimal moisture which is observed in many cases. This should be at least discussed.

Many soils have over 34% clay. I don't think they should be dismissed as unimportant. Also, what is the effect of silt? Equal to sand? This should be considered at least in discussions.

Some sections in Methods sound more like Intro.

I would be careful to insist on the function as being a mechanistic foundation. It is partially process based, and more than many approaches, but as any model, there are simplifications involved (the 'a' parameter for example).

The authors state that function is not too sensitive to the 'b' parameter. Some evidence would be welcome. Also 'b' is calculated as $n^2+1+\omega$. Where the 1 and omega come from were not clear to me, even with the supplementary material.

Logic and precision in the text should be checked throughout. Some example in page 2:

- 'processes controlling carbon (C) fluxes,'. Be more precise. What C flux?
- 'introducing significant uncertainty.' To what?
- 'soil respiration produces far more CO₂ annually than do anthropogenic fossil fuel emissions'. This statement if not put into context is misleading to the uninformed reader. This simple comparison is not valid.

A number of grammar mistakes need to be taken care of. The native speaking coauthors could easily improve the language.

Line numbers would make reviewing easier.

I don't think 'changed narrowly' is a valid expression.

Figures

generally way too small (also in supplementary)

Fig 1, get rid of ovals.

Fig 2. Split it in two figs. Also, change color and shape of microbe and SOC symbols to make them easier to recognize. Make color borders thicker. Explain better what the grey zones are and the figure in general, best in the main text itself. (Another reason to use article format. Cramming too much info in figures makes them unreadable.)

Fig 3. Too small.

Fig 4: separate into two figures.

Reviewer #2 (Remarks to the Author):

This manuscript deals with the response of heterotrophic soil respiration to changes in soil moisture content. Since soil respiration is an important flux with a significant impact on atmospheric CO₂ concentrations, it is important to understand how projected climate change (e.g. changes in temperature, water availability) will affect soil respiration. The temperature sensitivity of soil respiration has received considerable attention, but the effect of soil moisture has been studied less. Even the most mechanistic models describing soil respiration rely on empirical reduction functions to describe the impact of water content on soil respiration, and typically the parameters of these reduction functions are determined through calibration to site-specific data or left unchanged. In the current manuscript, a new approach to derive the moisture reduction function based on mechanistic considerations on substrate availability (dry end) and microbial respiration (wet end) is presented. The parameters of this novel moisture reduction function can be derived from readily available soil data, which opens up possibilities for meaningful regionalization and has the potential to considerably improve the physical realism in the representation of soil processes in earth system models.

The manuscript is well written and appropriately structured, so I have not comments in this direction. Although I am generally supportive of the intention of this manuscript, I cannot currently recommend this manuscript for publication. My main concerns are outlined in the four general comments listed below. I have also provided some more specific comments that the authors may want to consider.

1. The basic model concepts presented in this manuscript were already published in Yan et al. (2016). The novelty is in the application to multiple laboratory data sets (Figure 4), and the derived relationships to easily measurable soil properties. I think this is not emphasized enough in the current manuscript.

2. The authors claim to have developed a pore-scale model to describe the relevant processes associated with heterotrophic soil respiration. However, I fail to see why the developed model should be seen as a pore-scale model. In fact, concepts such as tortuosity and pore connectivity used to describe diffusion in the developed model are not required in pore scale models, since the pores are fully resolved in such models. Therefore, I would argue that the developed model is rooted in continuum-scale models that should be applied to so-called representative elementary volumes. Having said that, the spatial discretization used in the modelling was 20 micrometer. On this scale, I would argue that equations such as Eq. 9 and 10 are not valid.

3. Given the complexity of the model concept, I find that it is ideally suited for implementation in a fully mechanistic model describing water flow and gas transport in a fully resolved manner (Richards equation for water flow + gas distribution). However, Eq. (1) in the Methods seems to suggest that sources and sinks are not spatially resolved in vertical direction and that water content is assumed to be homogeneous within the soil profile. In my opinion, this is a strong simplification given the heterogeneous distribution of both water and organic matter in field soil. It would be insightful when the authors could discuss why this is their preferred modelling approach. Intuitively, I would argue that the uncertainty in soil respiration predictions introduced by neglecting vertical variation in water content, temperature, and organic matter is higher than the uncertainty introduced by the moisture reduction function.

4. The novel model concept has only been validated using laboratory (batch) experiments. For a high-impact journal such as Nature Communications, I think a validation using soil respiration measurements made in the field using chamber measurements should be attempted to make the study more convincing. Such a validation could also illustrate the improvement that can be achieved when using a more appropriate moisture reduction function. This could be followed by a concluding paragraph that puts the observed improvements of soil respiration predictions into the context of the projected climate change in order to provide a global context for the presented results.

SPECIFIC COMMENTS

Line 94. Please stress here that these are simulation results and not measurement results.

Line 148. Here it is suggested a dependence on soil structure is implied. Although this is potentially true, the associated parameters (n_1 , n_2 ,...) are fixed to single value in this study. Also, no suitable methods to estimate these parameters are available, so the suggested dependence on soil structure is rather weak here.

Line 160-163. This seems a rather unlikely explanation to me. Is it not more reasonable to argue that soils with higher clay content typically are more structured and form microaggregates where organic carbon is less accessible?

Line 226. I find the use of the word upscaling rather ambitious here. Where is this actually taking place?

Line 230-232. As argued above, vertical variation in water content and organic carbon content are also not considered. This should at least be acknowledged here.

Reviewers' comments:

Reviewer #1 (Remarks to the Author):

The paper by Yan et al. presents an interesting study describing a new function relating soil heterotrophic respiration with moisture content. The topic is of clear importance for modeling the carbon cycle in terrestrial ecosystems and contributions such as this have the potential to improve the predictive accuracy of general land surface models.

The dependency of the response of microbial activity to soil water content is indeed something that needs to be improved in soil C models. This study uses both model and data approaches to try to derive a generally applicable function that captures such dependencies. It applies some novel formulations, in particular by combining calculations of diffusivity with a parameter representing microbial-SOC collocation or with an oxygen supply restriction factor, depending on the case. The main message in the paper is that this new function is an improvement over previous functions as it correctly reproduces data obtained either from pore-scale simulations or lab observations.

Response: We thank the reviewer for the positive comments.

And here I get to the paper's main weak point. This is a study that is relatively heavy on equations, which are needed to understand what was done. A careful step by step description is required to go from the general problem to the solution. It is a complex topic that requires close examination. However, the paper was written in the letter format trying to very succinctly describe the work. I strongly believe that, while a short message can be delivered for this study (a new function was developed that improves over previous ones), the main part should consist of a clear explanation of the analysis itself and its implications, something that is incompletely described in the main text. This study should be about carefully explaining how and why the development was done and tested, something that requires a longer article format. The way it is now, the reader needs to shift between main text, methods and supplementary material in order to understand what is being described. The main text is not self-explanatory and does not read easily because of the above.

Response: In the revised manuscript, we addressed these comments by:

- 1) Adding background to emphasize the importance of mechanistic moisture functions (pages 2-3).**
- 2) Providing step-by-step descriptions in the derivation of the moisture function, f_m , in the Methods (pages 27-31), and describing the function more clearly in the main text (page 5).**
- 3) Explaining the assumption and simplifications used in the derivation of the function f_m (pages 27-31), and testing the assumptions using microscale simulations (pages 5-9).**
- 4) Evaluating the function, f_m , by using published data from different soil types (pages 9-11).**
- 5) Describing clearly how to use the function f_m in practical applications (pages 31-32), and testing its applicability and accuracy by comparing with laboratory and field observations (pages 11-12).**
- 6) Discussing the modeling results of the function f_m , and analyzing its advantages and weaknesses (pages 12-16).**

There are also a number of points that are not clear and need addressing.

In summary, I believe the study is very publishable and can become a significant contribution to the topic. However, the paper needs to be improved before publication, and will largely benefit from changing the format, probably leading to more citations afterwards.

Response: We thank the reviewer for the encouraging comments. This revision has been reformatted and expanded significantly.

Specific remarks:

It was not clear to me how the function was derived from the pore-scale model. It may be based on some of the same equations used for the pore scale model, such as the desorption or diffusion flux. But these are general equation, unless they were parameterized with the pore scale mode. What seems the case is that the function was validated against the pore scale model.

Response: The moisture function, f_m , utilized the same effective diffusion laws as in the microscale model. Notice the term “pore-scale model” was replaced by “microscale model” in the revised manuscript to more accurately describe the previously developed fine-scale model, in which continuum rules were used. The primary difference between the moisture function and the microscale model is that the former introduces unknown parameters while the latter requires fine-scale spatial input. We explicitly stated these connections and differences in the revised manuscript (page 3 lines 56-68). We now clearly present how the f_m was validated using microscale simulations (pages 5-9 lines 101-156) and was calibrated using published experimental data (pages 9-10 lines 157-170), and how the unknown parameters were quantified using measurable soil properties (pages 10-11 lines 171-183).

In several parts it was not clear if the pore-model results or the measured data was being referred to.

Response: The data sources are now clearly specified throughout this revision.

What happens to the part of $F_{total}[DOC]$ that does not diffuse to the microbes? This will add to a pool that should affect the flux.

Response: We neglected the impact of this part of DOC by assuming their concentration or amount is low in soils. The effect of this assumption was discussed in the revised manuscript (page 15 lines 274-278).

Oxygen flux: why is the oxygen gradient made to depend on SOC and not on the actual $[O_2]$ in the soil? What happens with deeper layers?

Response: Theoretically oxygen flux depends on actual $[O_2]$ in the soil, which is the starting point to derive the moisture function f_m under conditions of O_2 limit (Eq. 5 in the Methods). After a large number of simulations using simulated soil cores with different depths, porosities, organic C contents, and diffusion-related parameter values, we found that the gradient of $[O_2]$ at the soil-atmosphere

interface can be estimated using organic C content. This has been clarified in the revised manuscript (page 33 lines 636-639).

The effects of deeper layers were examined by using simulated soil cores with different depths, and the results showed that the function f_m is not very sensitive to the deep soil organic C. We discussed the implications of this finding in the revised manuscript (page 14 lines 248-252).

Not clear what ‘bottom of soils’ refers to. That is transfers from aerobic to anaerobic is also not clear.

Response: The sentence was rewritten in this revision (page 7 lines 123-124).

Why is it important to calculate θ_{op} ? It would be enough to express the function as the minimum of the substrate supply and oxygen supply functions.

Response: θ_{op} splits the soil respiration into two regimes with one limited by organic C bioavailability and the other by oxygen supply. In addition, the expression of the analytical θ_{op} provides a simple way to assess the effects of soil properties on soil HR-moisture relationships. In the revised manuscript, we stated and discussed these points (page 5 lines 97-98, pages 13-14 lines 241-260).

‘DOC and DO are both completely degraded’. This phrase is not clear, especially since DO has not been defined anywhere.

Response: Thank you for pointing this out. This sentence was rewritten (page 30 lines 608-610), and DO was defined (page 30 line 603).

The final function eq(11) depends on a and b. However, a Should distinguish between diffusion and diffusivity. Apart from the effects of texture on diffusivity introduced in eq(4), the concentration gradient and diffusive distance should influence the total flux. Even if these are implicitly dealt with in the function, they should be discussed. In this same context, recent previous works that deal with similar topics should be cited (Manzoni et al. 2016, Modeling coupled enzymatic and solute transport controls on decomposition in drying soils).

Response: Thank you for the suggestions. The impacts of concentration gradient and diffusive distance are now discussed (page 15 lines 274-278), and references including Manzoni et al. 2016 were added (page 15 line 278).

Given that a dependency with clay content is found, it would be appropriate to compare it to previous related finding, namely Moyano et al. 2012, already cited in the text.

Response: We now compare the results with previous findings, including Moyano’s work, and discuss the comparisons (pages 13-14 lines 241-260).

Data used to fit the relationship between ‘a’ and clay seems the same as used for validating the function. Independent data should be used. Related to this: the ‘wide range’ of published data used to test the function seems to be only 3 studies. Not a wide range of published data. Maybe of clay content. but it is not clear in the paper if the function actually performed well against independent observations over a wide range of soil types. In addition, it seems like the function cannot simulate a plateau of optimal moisture which is observed in many cases. This should be at least discussed.

Response: We thank the reviewer for these important comments, with which we agree. In the revised manuscript, we now:

- 1) Collect published data from numerous soil types (see Supplementary Table S2) to evaluate the derived function f_m (pages 9-10 lines 157-170), and establish the quantitative relationship between parameter a and clay content (pages 10-11 lines 171-183).**
- 2) Utilize independent laboratory and field measurement to assess the applicability and precision of the function f_m (pages 11-12 lines 184-201).**
- 3) Discuss the plateau around the optimal moisture observed in measurements (page 15 lines 283-288).**

Many soils have over 34% clay. I don’t think they should be dismissed as unimportant. Also, what is the effect of silt? Equal to sand? This should be considered at least in discussions. Some sections in Methods sound more like Intro.

Response: In the revised manuscript, soils with very low (< 0.016 g/g) or high (> 0.37 g/g) clay content were discussed (page 15 lines 281-283), as were the more general impacts of sand and silt (page 15 lines 278-281).

The Methods were revised by adding descriptions and assumptions to make these points clearer and easier to understand (pages 27-31).

I would be careful to insist on the function as being a mechanistic foundation. It is partially process based, and more than many approaches, but as any model, there are simplifications involved (the ‘a’ parameter for example).

Response: We have replaced “mechanistic” by “process-based” in this revision.

The authors state that function is not too sensitive to the ‘b’ parameter. Some evidence would be welcome. Also ‘b’ is calculated as $n_2 + 1 + \omega$. Where the 1 and ω come from were not clear to me, even with the supplementary material.

Response: The previous statement about the sensitivity of f_m to b has been deleted to avoid confusion. Instead, we discussed the impact of soil properties on the determination of b (page 13 lines 231-236). In the calculation of b using $1 + n_g + \omega$, 1 is from the pore area for O₂ diffusion at the soil-atmosphere

interface (Eq. 8 in Methods), ω is from the O₂ gradient at the soil-atmosphere interface (Eq. 7 in Methods). Now we revise the Methods to clarify these expressions (pages 29-30).

Logic and precision in the text should be checked throughout. Some example in page 2:

- ‘processes controlling carbon (C) fluxes,’. Be more precise. What C flux?

Response: We have clarified all such descriptions to make them more logical and precise.

- ‘introducing significant uncertainty.’ To what?

Response: This sentence was rewritten (page 2 line 26).

- ‘soil respiration produces far more CO₂ annually than do anthropogenic fossil fuel emissions’. This statement if not put into context is misleading to the uninformed reader. This simple comparison is not valid.

Response: This sentence was deleted. Instead, we emphasized the significance of moisture on soil heterotrophic respiration (page 2 lines 40-47).

A number of grammar mistakes need to be taken care of. The native speaking coauthors could easily improve the language.

Response: Our native co-authors have helped to correct the grammar mistakes.

Line numbers would make reviewing easier.

Response: Line numbers have been added.

I don’t think ‘changed narrowly’ is a valid expression.

Response: This expression was deleted.

Figures
generally way too small (also in supplementary)

Response: Figures were replotted and enlarged in this revision.

Fig 1, get rid of ovals.

Response: Fig. 1 was replotted (page 4).

Fig 2. Split it in two figs. Also, change color and shape of microbe and SOC symbols to make them easier to recognize. Make color borders thicker. Explain better what the grey zones are and the figure in general, best in the main text itself. (Another reason to use article format. Cramming too much info in figures makes them unreadable.)

Response: Thank you for the suggestions. All the suggestions were taken (page 6 Fig. 2). In addition, we removed the previous soil cores containing different porosities, ie., different gray zones, to present the results more clearly. This removal does not affect the results.

Fig 3. Too small.

Response: Enlarged (page 7 Fig. 3)

Fig 4: separate into two figures.

Response: Done (pages 9-10 Figs. 5-6)

Reviewer #2 (Remarks to the Author):

This manuscript deals with the response of heterotrophic soil respiration to changes in soil moisture content. Since soil respiration is an important flux with a significant impact on atmospheric CO₂ concentrations, it is important to understand how projected climate change (e.g. changes in temperature, water availability) will affect soil respiration. The temperature sensitivity of soil respiration has received considerable attention, but the effect of soil moisture has been studied less. Even the most mechanistic models describing soil respiration rely on empirical reduction functions to describe the impact of water content on soil respiration, and typically the parameters of these reduction functions are determined through calibration to site-specific data or left unchanged. In the current manuscript, a new approach to derive the moisture reduction function based on mechanistic considerations on substrate availability (dry end) and microbial respiration (wet end) is presented.

The parameters of this novel moisture reduction function can be derived from readily available soil data, which opens up possibilities for meaningful regionalization and has the potential to considerably improve the physical realism in the representation of soil processes in earth system models.

Response: We thank the reviewer for the positive comments.

The manuscript is well written and appropriately structured, so I have not comments in this direction. Although I am generally supportive of the intention of this manuscript, I cannot currently recommend this manuscript for publication. My main concerns are outlined in the four general comments listed below. I have also provided some more specific comments that the authors may want to consider.

1. The basic model concepts presented in this manuscript were already published in Yan et al. (2016). The novelty is in the application to multiple laboratory data sets (Figure 4), and the derived relationships to easily measurable soil properties. I think this is not emphasized enough in the current manuscript.

Response: This is a good point. In the revised manuscript, we now:

1) Explicitly state the relationship between the derived moisture function, f_m , and the previously published model in Yan et al. (2016) (page 3 lines 56-68)

2) Utilize numerous published data to calibrate the function f_m (pages 9-10 lines 157-170), and to establish the quantitative relationship between the function parameters and measurable soil properties (pages 10-11 lines 171-183). The good applicability of f_m and the quantification of parameters in f_m have been emphasized throughout the revised manuscript.

3) Provide a clear procedure how to apply the function f_m (pages 31-32 lines 620-632), and test it by comparing with laboratory and field measurements (pages 11-12 lines 184-201).

2. The authors claim to have developed a pore-scale model to describe the relevant processes associated with heterotrophic soil respiration. However, I fail to see why the developed model should be seen as a pore-scale model. In fact, concepts such as tortuosity and pore connectivity used to describe diffusion in the developed model are not required in pore scale models, since the pores are fully resolved in such models. Therefore, I would argue that the developed model is rooted in continuum-scale models that should be applied to so-called representative elementary volumes. Having said that, the spatial discretization used in the modelling was 20 micrometer. On this scale, I would argue that equations such as Eq. 9 and 10 are not valid.

Response: Thank you for the important comments. In the previous fine-scale model developed by Yan et al. (2016), the soil core includes three types of voxels: porous one without solid, solid one without pore, and mixed one with both pores and solids. This treatment considers the fact that soil texture such as clay is below the resolution used to discretize the soil core. The same treatment was used in this study. Therefore, Eqs. 9 and 10 are valid. In this revision, we changed the “pore-scale” to “microscale” to more precisely describe the fine-scale model, in which continuum rules were used.

3. Given the complexity of the model concept, I find that it is ideally suited for implementation in a fully mechanistic model describing water flow and gas transport in a fully resolved manner (Richards equation for water flow + gas distribution). However, Eq. (1) in the Methods seems to suggest that sources and sinks are not spatially resolved in vertical direction and that water content is assumed to be homogeneous within the soil profile. In my opinion, this is a strong simplification given the heterogeneous distribution of both water and organic matter in field soil. It would be insightful when the authors could discuss why this is their preferred modelling approach. Intuitively, I would argue that the uncertainty in soil respiration predictions introduced by neglecting vertical variation in water content, temperature, and organic matter is higher than the uncertainty introduced by the moisture reduction function.

Response: We agree with the reviewers about these factors. For example, the heterogeneous distributions of water, organic C, aggregates, etc., are undoubtedly important for the relationship

between soil heterotrophic respiration and water content. However, we exclude these heterogeneity in the derived moisture function f_m , because they are highly variable across soils and difficult to quantify. Instead, we discussed their impacts on the heterotrophic respiration-moisture relationship (page 13 lines 232-235, pages 14-15 lines 264-271). Furthermore, we discussed the effects of temperature and C input on the heterotrophic respiration-moisture relationships (page 15 lines 289-292).

4. The novel model concept has only been validated using laboratory (batch) experiments. For a high-impact journal such as Nature Communications, I think a validation using soil respiration measurements made in the field using chamber measurements should be attempted to make the study more convincing. Such a validation could also illustrate the improvement that can be achieved when using a more appropriate moisture reduction function. This could be followed by a concluding paragraph that puts the observed improvements of soil respiration predictions into the context of the projected climate change in order to provide a global context for the presented results.

Response: In the revised manuscript, we first validated the function f_m and quantified the function parameters using laboratory experiments (pages 9-10 lines 157-170). Then, the f_m was tested by comparing with both laboratory and field measurement to evaluate its applicability and accuracy (pages 10-11 lines 171-183). A paragraph was also added to compare the function f_m with commonly used empirical moisture functions (page 12 lines 202-209). In addition, the applications of f_m in Earth system models were discussed (page 16 lines 297-300).

SPECIFIC COMMENTS

Line 94. Please stress here that these are simulation results and not measurement results.

Response: The simulated and measured data were clearly referred throughout this revision. In addition, the source of data was highlighted at the beginning of each part (page 5 line 101, page 9 line 157, page 11 line 184).

Line 148. Here it is suggested a dependence on soil structure is implied. Although this is potentially true, the associated parameters (n_1, n_2, \dots) are fixed to single value in this study. Also, no suitable methods to estimate these parameters are available, so the suggested dependence on soil structure is rather weak here.

Response: We now explain and discuss how these diffusion-related parameters influence the function f_m , and cite the reference measuring their values for different textured soils (page 14 lines 252-256).

Line 160-163. This seems a rather unlikely explanation to me. Is it not more reasonable to argue that soils with higher clay content typically are more structured and form microaggregates where organic carbon is less accessible?

Response: In this revision, we added sentences to explain the impact of clay content on microbial accessibility to soil organic carbon (page 13 lines 228-231).

Line 226. I find the use of the word upscaling rather ambitious here. Where is this actually taking place?

Response: The word upscaling has been removed from this revision.

Line 230-232. As argued above, vertical variation in water content and organic carbon content are also not considered. This should at least be acknowledged here.

Response: The impact of heterogeneous water content and organic carbon content were discussed in this revision (page 13 lines 232-235, page 15 lines 268-269).

Reviewers' comments:

Reviewer #2 (Remarks to the Author):

I acted as reviewer 2 in the first round of reviews. If wanted, my reviewer comments and the associated reply by the author can be published together with the manuscript.

I have now evaluated the revised version of this manuscript, and I found that my main concerns were appropriately addressed. In particular, field data have been added for additional validation, and the presentation of the model and the associated assumptions is much clearer now. I found that the main manuscript is readable and understandable without going back and forth between the letter, methods, and supplementary information. However, the quality of writing can still be considerably improved, and I encourage the author team to take care of this. The specific comments below provide a non-exhaustive list of editorial issues that need attention, and some minor points that the authors may want to consider when preparing the final manuscript.

As already indicated in my original review, I am supportive of the idea of the manuscript. In the revision process, the manuscript has improved significantly. Therefore, I am able to recommend publication of this manuscript pending editorial improvements.

SPECIFIC COMMENTS

Line 46. What are “predictable models”? Not clear.

Line 57. “occurred” seems inappropriate here. Please improve writing.

Line 59. “nearly reproduced” is an awkward statement. Please improve.

Line 62-63. Not sure that this is what you want to say. The use of “limited to” seems incorrect to me here.

Line 90. What is “moisture degree”? Not clear.

Line 97-98. This was already stated above. No repetition required here.

Line 125. This formulation is unclear. Please improve.

Line 129-130. Not clear to me what you are trying to state here. What is now depending on soil heterogeneity?

Line 161-162. I am not so convinced that it is evident from these three panels that the performance is worse for water contents higher than the optimal water content.

Line 168. Use value instead of location here?

Line 177-183. The explanation of what the error bars represent should be improved. In both cases, they seem to ranges and not standard deviations. Please make this clearer.

Line 202. Should read: "As a final test...".

Line 209. Delete "do".

Line 217. Use climate change not changes.

Line 235. Here, I wonder how realistic incubation experiments are to represent oxygen supply in real soils. In my understanding, soil structure is destroyed before incubation, which should largely improve oxygen availability as compared to actual soil. This is well documented in Herbst et al. (2016) that is cited in the supplementary information. I think this warrants more discussion.

Line 240. Reconsider sentence. Perhaps "...compared well with measurements."

References. The list seems to contain duplicate references. Not sure that this is the desired format for Nature Communications.

Line 550. Throughout the manuscript, I would prefer the use of “soil physical” over “geophysical”. I associated the latter with the field of geophysics, which is not really covered in this manuscript.

Supplementary Figure 2. This figure is already included in the main manuscript. Not sure that it needs to be repeated here.

Supplementary Figure 5b. Here, you could plot a relation showing that a constant water-filled pore space describes the data reasonable well.

Reviewer #3 (Remarks to the Author):

Review of: “A moisture function of soil heterotrophic respiration incorporating microscale processes” by Yan et al.

Note: I wrote my review before reading the rebuttal letter to avoid being biased. My impression is that most of my comments ended up being complementary to those of the other reviewers, but one comment on the O₂ gradient overlaps with a previous criticism.

This work investigates the applicability of a model of respiration responses to changes in soil moisture to a range of soil types. The model is developed by considering C-limited and oxygen-limited regimes, and how the respiration rates scales with soil moisture under those conditions. Where oxygen and C are co-limiting, the respiration is maximized. This is conceptually similar to previous theoretical efforts (Skopp et al. 1990, Schjonning et al. 2003, Moyano et al. 2013, Manzoni et al. 2016), which are referred to in this manuscript (perhaps L224 is overstating a bit the novelty of this work). The main novelty is perhaps the interpretation of model parameters in light of pore-scale simulations with an existing model by the same group (Yan et al. 2016), leading to the interesting conclusion that microbial collocation is important to define the shape of the respiration curves. The results present various approaches for a step-by-step calibration and validation of the model. Texture dependent relationships are obtained to use the model across soils, and some evidence that

such a parameterization works better than previous ones is presented. Overall, this is an interesting work that complements ongoing activities in the area of soil heterotrophic respiration modelling. It attempts to develop a simple but physically sound relation for broad applications, and partly succeeds in this aim (“partly” because the validation dataset seems a bit limited compared to the calibration set, see Figure 7).

General comments

The model is based on the idea that C is ‘desorbed’ from particle surfaces, but how about enzymatic break-down, which is thought to be a key mechanism for release of C? I am not arguing against the current model formulation, but I wonder if the proposed model could capture enzymatic dynamics as well (albeit in an approximated way). The model also neglects DOC dynamics, assuming that the presence of DOC is never limiting for the desorption process (this limitation is acknowledged). This simplification has an important consequence – this model cannot account for DOC losses due to for example transport during drainage (leaching). While in the lab studies used for validation and calibration leaching is likely negligible (jars, not percolating soils), field conditions are different and DOC may be lost at high soil moisture. Perhaps this is why the data in Figure 7b indicate a large decrease in respiration just above the optimal soil moisture level.

The hypothesis presented in L581 is sound, but can it be tested with this model? In the previous paper by Yan et al. both C supply limitations and microbial inactivation had been considered. Now all these effects are lumped in the fitting exponent a , and in addition, percolation thresholds are neglected (while they were considered by Yan et al. (2016)). I am fine with lumping processes in model parameters for simplicity, but then it becomes harder to attribute patterns to processes. Figure 2 shows that collocation alone can make the respiration-moisture curves concave upwards, but these are results of a model. Are trends in the exponent a as estimated from fitting of data really due to collocation, or rather a combination of spatial heterogeneities, microbial community responses and other factors that have been lumped in a ?

While the points raised above are mainly ‘philosophical’ or model interpretation issues, I have some more technical concerns. The ΔO_2 term defined in Eq. 7 should – at least in my view – represent the overall transport of O_2 from the atmosphere to the bulk soil, not just the transport at the soil-atmosphere interface. This flux at the interface is not representative of O_2 conditions inside the sample. This can be a large gradient driving a locally large flux, while the majority of the soil could be O_2 depleted and exchanging O_2 very slowly with the surface. Moreover, the assumption that diffusion limits transport of both DOC and O_2 might be reasonable in dry soils, but in most cases diffusion is a very poor transport mechanism, and advective and dispersive mechanisms are predominant (water moves due to transpiration, redistribution, gravity...). This limitation might undermine the applicability of this model to soils except in lab conditions, but this issue is also openly discussed in this manuscript.

Other comments

L56: the cited paper is not the only one developed to study C and nutrient cycling at the pore scale (Ebrahimi and Or 2015, 2016).

L81-82: for simplicity and clarity I would change “low saturated” to “dry” and “high saturated” to “wet”.

L125: perhaps change to “fitted using the simulated data”?

L129: changed in b reflect the dynamics of oxygen in relation to C – in this simple model, the dynamics are collapsed on a single curve, but in reality depending on the time of sampling, the relation between respiration and soil moisture is probably different.

L169 and 182: how are these “certain ranges” defined?

L172: how was this relation derived? Using which data?

L228: changing exponent a could be due to shifting percolation threshold, which was however neglected here.

L236 and 600: b is an exponent, not a flux, so it cannot be interpreted as “O₂ supply” – it merely tells about the sensitivity of the O₂ supply to changes in air-filled porosity.

L272: diffusion is not the dominant mechanism at the soil core scale, unless water is perfectly still, which is typically not.

L533: missing year.

L597: how is k_{GO} determined? Does its numerical value matter?

L636: using which data? From simulations?

L641: “literature” (singular).

Comments on the supplementary materials:

Eq. 9-10: why neglecting the percolation thresholds? My impression is that having percolation thresholds depending on the soil heterogeneities or texture (which would be reasonable to expect) would lessen the effect of heterogeneity on the exponent a.

L72: some of these sentences are exactly as in the paper by Yan et al. (2016), Biogeochemistry

References

Ebrahimi, A., and D. Or. 2015. Hydration and diffusion processes shape microbial community organization and function in model soil aggregates. *Water Resources Research* 51:9804-9827.

Ebrahimi, A., and D. Or. 2016. Microbial community dynamics in soil aggregates shape biogeochemical gas fluxes from soil profiles - upscaling an aggregate biophysical model. *Global Change Biology* 22:3141-3156.

Manzoni, S., F. Moyano, T. Kätterer, and J. Schimel. 2016. Modeling coupled enzymatic and solute transport controls on decomposition in drying soils. *Soil Biology and Biochemistry* 95:275-287.

Moyano, F. E., S. Manzoni, and C. Chenu. 2013. Responses of soil heterotrophic respiration to moisture availability: An exploration of processes and models. *Soil Biology and Biochemistry* 59:72-85.

Schjonning, P., I. K. Thomsen, P. Moldrup, and B. T. Christensen. 2003. Linking soil microbial activity to water- and air-phase contents and diffusivities. *Soil Science Society of America Journal* 67:156-165.

Skopp, J., M. D. Jawson, and J. W. Doran. 1990. Steady-state aerobic microbial activity as a function of soil-water content. *Soil Science Society of America Journal* 54:1619-1625.

Yan, Z. F., C. X. Liu, K. E. Todd-Brown, Y. Y. Liu, B. Bond-Lamberty, and V. L. Bailey. 2016. Pore-scale investigation on the response of heterotrophic respiration to moisture conditions in heterogeneous soils. *Biogeochemistry* 131:121-134.

Reviewers' comments:

Reviewer #2 (Remarks to the Author):

I acted as reviewer 2 in the first round of reviews. If wanted, my reviewer comments and the associated reply by the author can be published together with the manuscript.

I have now evaluated this revised version of this manuscript, and I found that my main concerns were appropriately addressed. In particular, field data have been added for additional validation, and the presentation of the model and the associated assumptions is much clearer now. I found that the main manuscript is readable and understandable without going back and forth between the letter, methods, and supplementary information. However, the quality of writing can still be considerably improved, and I encourage the author team to take care of this. The specific comments below provide a non-exhaustive list of editorial issues that need attention, and some minor points that the authors may want to consider when preparing the final manuscript.

As already indicated in my original review, I am supportive of the idea of the manuscript. In the revision process, the manuscript has improved significantly. Therefore, I am able to recommend publication of this manuscript pending editorial improvements.

Responses: We thank reviewer #2 for the positive comments. We have worked hard to improve the quality of writing throughout. The responses to specific comments and minor points are presented as below.

SPECIFIC COMMENTS

Line 46. What are “predictable models”? Not clear.

Responses: “predictable models” is now replaced by the clearer phrase “mechanistic models to predict the response of HR rates to moisture change” (page 2 line 44)

Line 57. “occurred” seems inappropriate here. Please improve writing.

Responses: The sentence has been rewritten (page 3 lines 59-60)

Line 59. “nearly reproduced” is an awkward statement. Please improve.

Responses: “nearly reproduced” is now changed to “generated HR-moisture relationships in agreement with measurements” (page 3 lines 60-61)

Line 62-63. Not sure that this is what you want to say. The use of “limited to” seems incorrect to me here.

Responses: The sentence has been changed to “preventing its application in large-scale modeling of soil C decomposition” (page 3 line 63)

Line 90. What is “moisture degree”? Not clear.

Responses: “moisture degree” is now changed to “water content” (page 5 line 88)

Line 97-98. This was already stated above. No repetition required here.

Responses: Deleted

Line 125. This formulation is unclear. Please improve.

Responses: The sentence has been rewritten to clarify the formulation (pages 6-7 lines 121-122), and a statement is added to explain how to get the formulation (page 8 lines 139-140).

Line 129-130. Not clear to me what you are trying to state here. What is now depending on soil heterogeneity?

Responses: This statement is deleted to avoid confusion. Instead, we discuss the effect of soil characteristics on the value of b in this revised manuscript (pages 13-14 lines 242-248).

Line 161-162. I am not so convinced that it is evident from these three panels that the performance is worse for water contents higher than the optimal water content.

Responses: This sentence has been rewritten (pages 9 lines 159-160), and figures are added to show the comparisons between f_m and measured HR-moisture relationships for more soil types (Supplementary Fig. 6).

Line 168. Use value instead of location here?

Responses: Changed (page 9 lines 165)

Line 177-183. The explanation of what the error bars represent should be improved. In both cases, they seem to ranges and not standard deviations. Please make this clearer.

Responses: These sentences have been rewritten to better explain the bars (pages 10-11 lines 177-180).

Line 202. Should read: “As a final test...”.

Responses: Changed (page 12 line 198)

Line 209. Delete “do”.

Responses: Deleted

Line 217. Use climate change not changes.

Responses: Changed (pages 12 lines 215)

Line 235. Here, I wonder how realistic incubation experiments are to represent oxygen supply in real soils. In my understanding, soil structure is destroyed before incubation, which should largely improve oxygen availability as compared to actual soil. This is well documented in Herbst et al. (2016) that is cited in the supplementary information. I think this warrants more discussion.

Responses: We now discuss the effects of real soils (e.g., aggregates, macro-pores, and fractures) on oxygen supply in this revision (pages 16-17 lines 317-321).

Line 240. Reconsider sentence. Perhaps “...compared well with measurements.”

Responses: Changed (pages 14 line 258).

References. The list seems to contain duplicate references. Not sure that this is the desired format for Nature Communications.

Responses: Thanks for catching this. The duplicate references are deleted in this revision.

Line 550. Throughout the manuscript, I would prefer the use of “soil physical” over “geophysical”. I associated the latter with the field of geophysics, which is not really covered in this manuscript.

Responses: Changed (page 17 line 339).

Supplementary Figure 2. This figure is already included in the main manuscript. Not sure that it needs to be repeated here.

Responses: The simulated soil cores used in supplementary Figure 2 have different depths, porosity values, and organic C contents as that in Fig. 2. We now clarify the differences and replot supplementary Figure 2 in this revision (page 6 lines 105-106, supplementary Figure 2).

Supplementary Figure 5b. Here, you could plot a relation showing that a constant water-filled pore space describes the data reasonable well.

Responses: A plot showing the relationship between soil HR rates and a constant water-filled pore space is added in this revision (supplementary Figure 5b).

Reviewer #3 (Remarks to the Author):

Review of: “A moisture function of soil heterotrophic respiration incorporating microscale processes” by Yan et al.

Note: I wrote my review before reading the rebuttal letter to avoid being biased. My impression is that most of my comments ended up being complementary to those of the other reviewers, but one comment on the O₂ gradient overlaps with a previous criticism.

This work investigates the applicability of a model of respiration responses to changes in soil moisture to a range of soil types. The model is developed by considering C-limited and oxygen-limited regimes, and how the respiration rates scales with soil moisture under those conditions. Where oxygen and C are co-limiting, the respiration is maximized. This is conceptually similar to previous theoretical efforts (Skopp et al. 1990, Schjonning et al. 2003, Moyano et al. 2013, Manzoni et al. 2016), which are referred to in this manuscript (perhaps L224 is overstating a bit the novelty of this work). The main novelty is perhaps the interpretation of model parameters in light of pore-scale simulations with an existing model by the same group (Yan et al. 2016), leading to the interesting conclusion that microbial collocation is important to define the shape of the respiration curves. The results present various approaches for a step-by-step calibration and validation of the model.

Texture dependent relationships are obtained to use the model across soils, and some evidence that such a parameterization works better than previous ones is presented. Overall, this is an interesting

work that complements ongoing activities in the area of soil heterotrophic respiration modelling. It attempts to develop a simple but physically sound relation for broad applications, and partly succeeds in this aim (“partly” because the validation dataset seems a bit limited compared to the calibration set, see Figure 7).

Responses: We thank the reviewer for the critical comments. In this revised manuscript, we acknowledge the previous theoretical efforts, and emphasize the novelty of this work (page 3 lines 51-53, page 12 lines 215-219). The responses to comments on O₂ gradient and others are described as below.

General comments

The model is based on the idea that C is ‘desorbed’ from particle surfaces, but how about enzymatic break-down, which is thought to be a key mechanism for release of C? I am not arguing against the current model formulation, but I wonder if the proposed model could capture enzymatic dynamics as well (albeit in an approximated way). The model also neglects DOC dynamics, assuming that the presence of DOC is never limiting for the desorption process (this limitation is acknowledged). This simplification has an important consequence – this model cannot account for DOC losses due to for example transport during drainage (leaching). While in the lab studies used for validation and calibration leaching is likely negligible (jars, not percolating soils), field conditions are different and DOC may be lost at high soil moisture. Perhaps this is why the data in Figure 7b indicate a large decrease in respiration just above the optimal soil moisture level.

Responses: In this revised manuscript, we acknowledge the importance of enzyme in decomposing soil organic carbon (page 13 lines 234-235). However, we do not include the enzymatic dynamics in the moisture function f_m , because we aim to develop a feasible model that can be easily used for different soil types across spatial scales. Including the enzymatic dynamics would complicate the formulation and applications of f_m . Instead, we discuss the possibility of studying the enzymatic dynamics using microscale modeling and then upscaling to macroscale models (page 13 lines 231-237).

We neglected the DOC dynamics mainly because they are related to hydrological conditions and are difficult to express mathematically in the moisture function f_m . In this revised manuscript, we discuss the effects of advective and dispersive transport caused by water movement (precipitation, drainage, and plant transpiration) in field sites (page 16 lines 313-316).

The hypothesis presented in L581 is sound, but can it be tested with this model? In the previous paper by Yan et al. both C supply limitations and microbial inactivation had been considered. Now all these effects are lumped in the fitting exponent a , and in addition, percolation thresholds are neglected (while they were considered by Yan et al. (2016)). I am fine with lumping processes in

model parameters for simplicity, but then it becomes harder to attribute patterns to processes. Figure 2 shows that collocation alone can make the respiration-moisture curves concave upwards, but these are results of a model. Are trends in the exponent a as estimated from fitting of data really due to collocation, or rather a combination of spatial heterogeneities, microbial community responses and other factors that have been lumped in a?

Responses: Actually, the hypothesis has been tested using the microscale model. Fig. 2 shows that the relationship between soil HR rates and water content can be described by the SOC-microorganism collocation factor a . In this revised manuscript, we clearly state how the hypothesis is tested using microscale modeling (page 5 lines 99-101).

The microbial inactivation and percolation thresholds are neglected mainly because we want to derive a simple and feasible moisture function. The microbial inactivation becomes important under only very dry conditions, and thus has only a minor effect on the HR-moisture relationship under most moisture conditions; the percolation thresholds primarily reduce the absolute rather than relative HR rates, thus having slight impact on the HR-moisture relationship when the water or air content is above the percolation thresholds. In this revised manuscript, we discuss these effects (page 16 lines 295-298, page 15 lines 288-292).

We agree with the reviewer that other factors may cause the change in the respiration-moisture curves in natural soils. We now provide caveats about using modeling results to explain experimental observations (page 13 lines 228-231).

While the points raised above are mainly ‘philosophical’ or model interpretation issues, I have some more technical concerns. The Δ_{O_2} term defined in Eq. 7 should – at least in my view – represent the overall transport of O₂ from the atmosphere to the bulk soil, not just the transport at the soil-atmosphere interface. This flux at the interface is not representative of O₂ conditions inside the sample. This can be a large gradient driving a locally large flux, while the majority of the soil could be O₂ depleted and exchanging O₂ very slowly with the surface. Moreover, the assumption that diffusion limits transport of both DOC and O₂ might be reasonable in dry soils, but in most cases diffusion is a very poor transport mechanism, and advective and dispersive mechanisms are predominant (water moves due to transpiration, redistribution, gravity...). This limitation might undermine the applicability of this model to soils except in lab conditions, but this issue is also openly discussed in this manuscript.

Responses: The ∇_{O_2} in Eq. 7 is the gradient of gaseous O₂ concentrations between the top soil surface and the atmosphere, and Eq. 5 represents the overall diffusion rate of O₂ from the atmosphere to the bulk soil when ignoring the diffusion through water phase (page 19 lines 376-378). We agree with the reviewer that the overall transport of O₂ from the atmosphere can not represent the O₂ conditions inside soils, and argue this is one important reason why the O₂ supply restriction factor, b , is not mathematically related to soil properties. We now

discuss the effects of soil heterogeneity, including spatial distributions of soil structure, texture, and water content, on the supply of O₂ (pages 13-14 lines 242-248, pages 16-17 lines 317-321). We also discuss the effect of advective and dispersive mechanisms in transporting DOC and O₂ under certain conditions in field sites (page 16 lines 313-316).

Other comments

L56: the cited paper is not the only one developed to study C and nutrient cycling at the pore scale (Ebrahimi and Or 2015, 2016).

Responses: A sentence has been added to introduce more works related to mechanistic models (page 3 lines 53-55).

L81-82: for simplicity and clarity I would change “low saturated” to “dry” and “high saturated” to “wet”.

Responses: Changed (page 4 line 79)

L125: perhaps change to “fitted using the simulated data”?

Responses: The sentence is rewritten (pages 6-7 lines 121-122).

L129: changed in b reflect the dynamics of oxygen in relation to C – in this simple model, the dynamics are collapsed on a single curve, but in reality depending on the time of sampling, the relation between respiration and soil moisture is probably different.

Responses: We now discuss the effect of timing on soil HR-moisture relationships in this revision (page 17 lines 322-324).

L169 and 182: how are these “certain ranges” defined?

Responses: “certain ranges” is changed to “a range” (page 9 line 165)

L172: how was this relation derived? Using which data?

Responses: The data and methods related to this relation have been added (page 10 line 169, page 11 line 180)

L228: changing exponent a could be due to shifting percolation threshold, which was however neglected here.

Responses: The effect of percolation thresholds is now discussed (page 15 lines 288-292).

L236 and 600: b is an exponent, not a flux, so it cannot be interpreted as “O₂ supply” – it merely tells about the sensitivity of the O₂ supply to changes in air-filled porosity.

Responses: The statements are rewritten in this revision (page 14 line 250, page 20 lines 391-392).

L272: diffusion is not the dominant mechanism at the soil core scale, unless water is perfectly still, which is typically not.

Responses: The effects of advective and dispersive transport are now discussed (page 16 lines 313-316)

L533: missing year.

Responses: Added

L597: how is k_{GO} determined? Does its numerical value matter?

Responses: k_{GO} was estimated by the intercepts of $\log(\nabla_{O_2}) - \log(\phi - \theta)$ curves with y-coordinate (Fig. 3a). Its value affects the supply rate of O₂ from the atmosphere. We now clarify the estimation of k_{GO} in this revision (page 21 lines 413-416).

L636: using which data? From simulations?

Responses: The data are from simulations by the microscale model. The statements are now rewritten to clarify these information (page 21 lines 414-416).

L641: “literature” (singular).

Responses: corrected

Comments on the supplementary materials:

Eq. 9-10: why neglecting the percolation thresholds? My impression is that having percolation thresholds depending on the soil heterogeneities or texture (which would be reasonable to expect) would lessen the effect of heterogeneity on the exponent a .

Responses: We neglect the percolation thresholds, because they have a relatively small effect on the relative HR rates that were used in the HR-moisture relationship, although they reduce the absolute HR rates. This negligence simplifies the formulation and application of function f_m . In this revised manuscript, we now discuss the impact of the percolation thresholds (page 15 lines 288-292).

L72: some of these sentences are exactly as in the paper by Yan et al. (2016), *Biogeochemistry*

Responses: These sentences are now changed (page 4 lines 74-75 in the supplementary materials).

References

- Ebrahimi, A., and D. Or. 2015. Hydration and diffusion processes shape microbial community organization and function in model soil aggregates. *Water Resources Research* 51:9804-9827.
- Ebrahimi, A., and D. Or. 2016. Microbial community dynamics in soil aggregates shape biogeochemical gas fluxes from soil profiles - upscaling an aggregate biophysical model. *Global Change Biology* 22:3141-3156.
- Manzoni, S., F. Moyano, T. Kätterer, and J. Schimel. 2016. Modeling coupled enzymatic and solute transport controls on decomposition in drying soils. *Soil Biology and Biochemistry* 95:275-287.
- Moyano, F. E., S. Manzoni, and C. Chenu. 2013. Responses of soil heterotrophic respiration to moisture availability: An exploration of processes and models. *Soil Biology and Biochemistry* 59:72-85.
- Schjonning, P., I. K. Thomsen, P. Moldrup, and B. T. Christensen. 2003. Linking soil microbial activity to water- and air-phase contents and diffusivities. *Soil Science Society of America Journal* 67:156-165.
- Skopp, J., M. D. Jawson, and J. W. Doran. 1990. Steady-state aerobic microbial activity as a function of soil-water content. *Soil Science Society of America Journal* 54:1619-1625.
- Yan, Z. F., C. X. Liu, K. E. Todd-Brown, Y. Y. Liu, B. Bond-Lamberty, and V. L. Bailey. 2016. Pore-scale investigation on the response of heterotrophic respiration to moisture conditions in heterogeneous soils. *Biogeochemistry* 131:121-134.

REVIEWERS' COMMENTS:

Reviewer #2 (Remarks to the Author):

I acted as reviewer 2 in the first round of reviews. Although I was already satisfied with the changes made after the first revision, I was contacted by the editor to evaluate this second revised version and the associated reply to the reviewer comment. I have accepted this request, and I found that the review process was fruitful, and that the final manuscript has been considerably improved by the (partly) critical but constructive reviewer comments. In my reading of the reviewer comments and the associated replies, there is perhaps a disagreement on how to simplify the known mechanistic processes that affect heterotrophic respiration so that a simplified model is obtained that can easily be parameterized using basic soil information. The simplifying assumptions with respect to oxygen supply are perhaps appropriate for laboratory incubation experiments with disturbed soil, but may be less appropriate for field measurements. This concern has been expressed in direct and indirect ways in both round of reviews, and has been appropriately recognized by the authors in the two revisions. Similarly, the importance of advective and dispersive transport of DOC has now been appropriately discussed. Therefore, I recommend to accept this manuscript for publication now.

REVIEWERS' COMMENTS:

Reviewer #2 (Remarks to the Author):

I acted as reviewer 2 in the first round of reviews. Although I was already satisfied with the changes made after the first revision, I was contacted by the editor to evaluate this second revised version and the associated reply to the reviewer comment. I have accepted this request, and I found that the review process was fruitful, and that the final manuscript has been considerably improved by the (partly) critical but constructive reviewer comments. In my reading of the reviewer comments and the associated replies, there is perhaps a disagreement on how to simplify the known mechanistic processes that affect heterotrophic respiration so that a simplified model is obtained that can easily be parameterized using basic soil information. The simplifying assumptions with respect to oxygen supply are perhaps appropriate for laboratory incubation experiments with disturbed soil, but may be less appropriate for field measurements. This concern has been expressed in direct and indirect ways in both round of reviews, and has been appropriately recognized by the authors in the two revisions. Similarly, the importance of advective and dispersive transport of DOC has now been appropriately discussed. Therefore, I recommend to accept this manuscript for publication now.

Responses: We appreciate the reviewer 2 for spending time reviewing the second revised version, and thank the reviewer 2 for supporting publishing our work in Nature Communications.